# DualCnst: Enhancing Zero-Shot Out-of-Distribution Detection via Text-Image Consistency in Vision-Language Models

**Fayi Le**[1]  **Wenwu He**[1,2†]  **Chentao Cao**[3]  **Dong Liang**[4†]  **Zhuo-Xu Cui**[4†]

[1]School of Computer Science and Mathematics, Fujian University of Technology
[2]Fujian Provincial Key Laboratory of Big Data Mining and Applications
[3]Department of Computer Science, Hong Kong Baptist University
[4]Shenzhen Institutes of Advanced Technology, Chinese Academy of Sciences
`fayi@smail.fjut.edu.cn`
`hwwhbb@163.com, chentaocao1224@gmail.com`
`{dong.liang, zx.cui}@siat.ac.cn`

## Abstract

Pretrained vision-language models (VLMs), such as CLIP, have shown promising zero-shot out-of-distribution (OOD) detection capabilities by leveraging semantic similarities between input images and textual labels. However, most existing approaches focus solely on expanding the label space in the text domain, ignoring complementary visual cues that can further enhance discriminative power. In this paper, we introduce DualCnst, a novel framework that integrates text-image dual consistency for improved zero-shot OOD detection. Specifically, we generate synthetic images from both ID and mined OOD textual labels using a text-to-image generative model, and jointly evaluate each test image based on (i) its semantic similarity to class labels and (ii) its visual similarity to the synthesized images. The resulting unified score function effectively combines multimodal information without requiring access to in-distribution images or additional training. We further provide theoretical analysis showing that incorporating multimodal negative labels reduces score variance and improves OOD separability. Extensive experiments across diverse OOD benchmarks demonstrate that DualCnst achieves state-of-the-art performance while remaining scalable, data-agnostic, and fully compatible with prior text-only VLM-based methods. The code is publicly available at: `https://github.com/TMLSIAT/DualCnst`.

## 1 Introduction

Out-of-distribution (OOD) detection aims to identify whether an input sample lies outside the training data distribution of a machine learning model during inference [1]. This capability is critical for preventing erroneous predictions when models encounter novel or anomalous inputs. It is especially vital in high-stakes applications such as medical imaging [2, 3, 4] and autonomous driving [5, 6, 7] where failures in OOD detection may lead to severe consequences like misdiagnoses or safety hazards.

Traditional OOD detection methods primarily rely on representations learned from the image modality. While effective to some extent, such approaches often neglect the rich semantic structure embedded in textual descriptions. The emergence of large-scale vision-language models (VLMs), such as CLIP [8],

---

[†]Corresponding authors: Wenwu He (`hwwhbb@163.com`), Dong Liang (`dong.liang@siat.ac.cn`), Zhuo-Xu Cui (`zx.cui@siat.ac.cn`)

39th Conference on Neural Information Processing Systems (NeurIPS 2025).

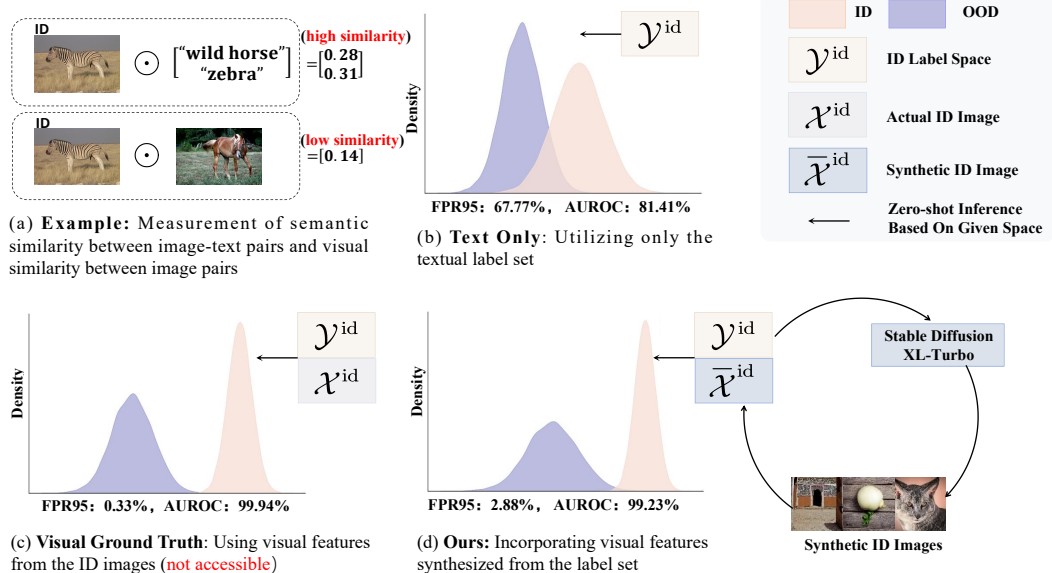

Figure 1: Motivating the use of visual similarity in OOD detection. (a) Semantically similar but visually distinct classes (e.g., wild horses and zebras) present a major challenge for text-only OOD detection. (b) Using only semantic similarity (e.g., CLIP text embeddings) results in significant overlap between ID and OOD score distributions. (c) Incorporating visual features from real ID images helps distinguish these ambiguous OOD samples by better capturing visual structure. (d) Even without access to real ID images, synthesizing class-conditional image exemplars via text-to-image models enables similar gains, narrowing the gap between ID and OOD scores. CIFAR-100 [10] is used as the ID dataset, and iNaturalist [11] as the OOD dataset.

has enabled the alignment of visual and textual modalities in a shared embedding space, thereby facilitating zero-shot OOD detection by measuring the similarity between test images and class names in natural language. For example, methods like MCM [9] perform inference by computing cosine similarity between test image embeddings and textual label embeddings, classifying inputs with low similarity scores as OOD—without requiring any additional training.

However, relying solely on semantic similarity has inherent limitations. Challenging OOD examples—particularly those lying near the decision boundary—may share semantic attributes with in-distribution (ID) classes, thereby causing significant overlap between ID and OOD score distributions (Figure 1(b)). Interestingly, such samples often remain visually distinguishable despite semantic ambiguity. For example, "zebra" and "wild horse" share strong semantic overlap but are visually distinct due to the zebra's unique striping (1(a)). This observation motivates a key question:

*Can visual similarity between test samples and ID/OOD exemplars improve the separability of semantically similar but visually distinct samples?*

We empirically confirm this hypothesis in Figure 1(c), where access to actual ID images allows more effective separation of ambiguous OOD cases. Unfortunately, real ID images are typically unavailable in open-world settings due to privacy, licensing, or infrastructure constraints. To overcome this limitation, we propose synthesizing visual exemplars using text-to-image diffusion models. As shown in Figure 1(d), these generated samples can approximate the visual structure of class concepts, enabling reliable similarity computation even without access to real ID data.

Building on this idea, we propose DualCnst, a novel zero-shot OOD detection framework that integrates semantic and visual consistency across the label space. Specifically, we synthesize images for both ID and mined OOD labels and define a scoring function that jointly considers: (i) semantic similarity between a test image and textual labels, and (ii) visual similarity between the test image and synthesized images. DualCnst offers several key advantages: (1) *Training-free*: It does not require fine-tuning or any labeled ID data. (2) *Data-agnostic*: It generalizes across domains without assuming access to ID images. (3) *Scalable*: Its modular similarity function can be plugged into existing VLM-based pipelines with minimal overhead.

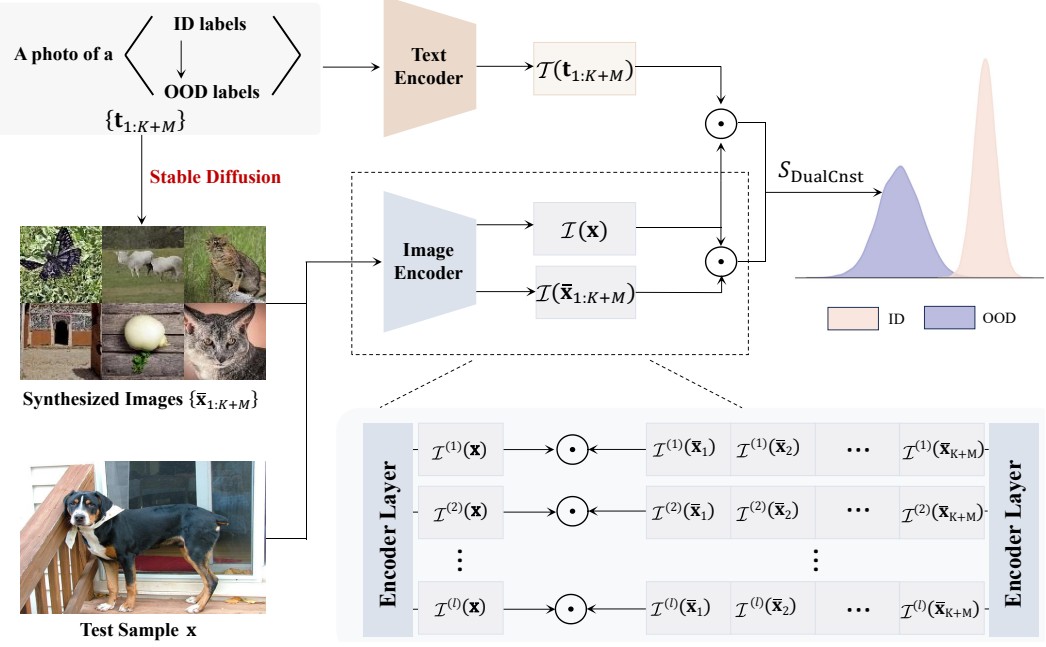

Figure 2: The framework of the proposed DualCnst is outlined as follows. Given a set of ID class labels $\mathcal{Y}^{id}$, we first leverage NegLabel [12]) to generate OOD labels $\mathcal{Y}^{ood}$. These class labels are then input into Stable Diffusion [13] to synthesize both ID and OOD images. Subsequently, both the ID/OOD class labels and the synthesized images are fed into the text and image encoders to construct the textual and image classifiers. During the testing phase, given an input image, its visual features are extracted using the image encoder, and the semantic similarity with the class labels is computed, along with the visual similarity to the synthesized images. Finally, the OOD score is derived by scaling and coupling these similarities using the proposed detection score function $S_{\text{DualCnst}}$.

Our contributions can be summarized as follows:

- We propose DualCnst, a dual-consistency framework that fuses semantic and visual similarities for zero-shot OOD detection (Section 3).

- We introduce a text-driven synthetic image generation pipeline to build visual exemplars for both ID and OOD labels without accessing in-distribution samples (Section 3).

- We provide theoretical analysis showing that multimodal negative labels reduce similarity score variance, enhancing ID/OOD separability (Section 4).

- Extensive experiments on standard OOD benchmarks show that DualCnst consistently outperforms previous zero-shot approaches. In particular, it achieves absolute gains of 3.95%, 3.9%, 9.9% in FPR95 on ImageNet-1K far-OOD, near-OOD, and robust-OOD tasks, respectively (Section 5).

## 2 Preliminaries

**CLIP and Zero-shot OOD Detection:** CLIP [8] is a multimodal pre-trained model designed to align visual and textual modalities within a shared embedding space. Trained on large-scale image-text datasets using contrastive learning, CLIP consists of an image encoder and a text encoder that generate embeddings for images and text, respectively. By computing cosine similarity between these embeddings, the model performs similarity-based matching. A key strength of CLIP is its remarkable zero-shot capability: trained on diverse and extensive image-text pairs, it can be directly applied to various vision tasks—including image classification [14, 15, 16, 17], object detection [18, 19, 20], semantic segmentation [21, 22, 23], and OOD detection—without requiring additional labeled data or fine-tuning.

For zero-shot OOD detection, CLIP determines whether an input image belongs to one of the known categories or represents an OOD sample. This is achieved by comparing the image's visual features

with the semantic representations of known class labels encoded as text. Images with low similarity to all known labels are identified as OOD samples. This zero-shot paradigm offers high flexibility, allowing CLIP to generalize across diverse domains without retraining, making it a powerful tool for OOD detection in real-world applications.

**Stable Diffusion:** Stable Diffusion is a generative model based on Latent Diffusion Models (LDMs) [13], designed for efficient text-to-image synthesis. Unlike conventional diffusion models that operate in pixel space, Stable Diffusion performs the diffusion process in a lower-dimensional latent space, significantly enhancing computational efficiency and scalability. The model employs a pre-trained Variational Autoencoder (VAE) [24] to encode high-resolution images into a compact latent representation, which serves as the input for the diffusion process. Within this latent space, a U-Net-based [25] denoising network executes both forward and reverse diffusion: in the forward process, noise is gradually added to the latent representation until it converges to a Gaussian distribution, while in the reverse process, the model learns to iteratively denoise the latent representation, reconstructing it into the original data distribution.

To enhance the fidelity and semantic alignment of generated images, Stable Diffusion incorporates CLIP as a guidance mechanism during the reverse diffusion process [26]. CLIP provides a similarity-based gradient signal that directs the latent representation toward alignment with the textual prompt, ensuring that the generated images faithfully capture both the semantic intent and fine-grained details. This method builds on previous CLIP-guided generative models [27, 28], which utilize multimodal representations to improve the coherence and expressiveness of generated content [29]. By leveraging CLIP's semantic understanding, Stable Diffusion generates visually coherent and contextually relevant images, even for abstract or complex prompts. This significantly broadens the model's applicability in text-to-image synthesis [28].

# 3  Text-Image Dual Consistency-Guided OOD Detection

In this paper, a novel approach is proposed to enhance zero-shot OOD detection performance by leveraging text-image dual consistency. Specifically, the method is divided into two stages: (i) Synthesis Stage: To evaluate the visual similarity of test samples with ID and OOD images, a text-to-image generative model, Stable Diffusion, is employed to synthesize image labels from the combined label space, $\mathcal{Y}^{id} \cup \mathcal{Y}^{ood}$. (ii) Testing Stage: To integrate textual and visual information, a novel score function is proposed. This function simultaneously evaluates the semantic similarity between test images and textual labels and measures the visual similarity between test samples and the synthesized ID/OOD image labels. The overall framework of the proposed method is illustrated in Figure 2.

## 3.1  Synthesize Images from the Label Space

To broaden the scope of visual information, NegLabel [12] is employed to identify potential OOD labels, which serve as prompts for an image generator. These prompts guide the generation of semantically consistent visual representations for OOD images. The label space is defined as $\mathcal{Y}^{id} \cup \mathcal{Y}^{ood} = y_1, y_2, \ldots, y_K, y_{K+1}, \ldots, y_{K+M}$, where $K$ denotes the number of ID labels and $M$ denotes the number of OOD labels.

To ensure semantic alignment between textual descriptions and generated images, the diffusion model's capacity for aligning textual and visual representations is utilized. For each label, a consistent text prompt, such as "A photo of a <label>," is constructed. These prompts are input into the diffusion model to generate synthetic images semantically aligned with the combined label space $\mathcal{Y}^{id} \cup \mathcal{Y}^{ood}$. This process enriches visual information and addresses the limitations of relying solely on semantic information for image-text alignment.

The generated images are represented as $\bar{\mathcal{X}} = \{\bar{\mathbf{x}}_i\}$, where each $\bar{\mathbf{x}}_i$ corresponds to a unique synthetic image associated with a specific label. These images not only capture the known ID data distributions but also simulate visual representations of OOD categories. By integrating this diverse set of synthetic images into the OOD detection process, the proposed method enhances the model's ability to differentiate ID from OOD instances. This is achieved by leveraging visual distinctions between ID and OOD images, leading to more accurate identification and rejection of OOD samples.

## 3.2 Integrate Textual and Visual Metrics for OOD Detection

We calculate the visual similarity between the test sample $\mathbf{x}$ and the synthesized image set $\bar{\mathcal{X}}$, as well as the semantic similarity with the label set $\mathcal{Y}$, in the feature space encoded by CLIP's text encoder $\mathcal{T}(\cdot)$ and image encoder $\mathcal{I}(\cdot)$.

**Image-to-Image Similarity.** In particular, both low-level and high-level visual features are incorporated. We extract features from intermediate layers and the final output layer of the image encoder to calculate cosine similarity between the test sample and the synthetic images at multiple levels of representation. Distinct weights are assigned to each layer to balance their contributions. For instance, using ViT-B/16 as the visual encoder, we select the third, sixth, ninth, and final semantic layers to compute cosine similarity between the test image and each synthetic image. A weight of $0.25$ is assigned to the similarity score from each layer, and the overall visual similarity is calculated as the weighted sum of these scores.

The visual similarity between the input image $\mathbf{x}$ and the synthesized image set $\bar{\mathcal{X}}$ is defined as:

$$s_{i,\text{img}}^{(l)}(\mathbf{x}) = \frac{\mathcal{I}^{(l)}(\mathbf{x}) \cdot \mathcal{I}^{(l)}(\bar{\mathbf{x}}_i)}{\|\mathcal{I}^{(l)}(\mathbf{x})\| \cdot \|\mathcal{I}^{(l)}(\bar{\mathbf{x}}_i)\|}; \quad \bar{\mathbf{x}}_i \in \bar{\mathcal{X}}. \tag{1}$$

where $\mathcal{I}^{(l)}(\mathbf{x})$ represents the feature embedding at layer $l$. The final similarity score $s_{i,\text{img}}(\mathbf{x})$ is obtained by summing the weighted similarity scores across all layers:

$$s_{i,\text{img}}(\mathbf{x}) = \sum_{l=1}^{L} w_l \cdot s_{i,\text{img}}^{(l)}(\mathbf{x}), \tag{2}$$

where $w_l$ represents the weight assigned to layer $l$ and is defined as:

$$w_l = \begin{cases} r, & l < L \\ 1 - r \cdot (L-1), & l = L \end{cases},$$

where $L$ denotes the total number of layers in the visual encoder, and $r$ is the weight factor applied to intermediate layers, ensuring a balanced contribution across all layers.

**Image-to-Text Similarity.** The semantic similarity between the test image $\mathbf{x}$ and the combined label space $\mathcal{Y}^{\text{id}} \cup \mathcal{Y}^{\text{ood}}$ is computed as:

$$s_{i,\text{text}}(\mathbf{x}) = \frac{\mathcal{I}(\mathbf{x}) \cdot \mathcal{T}(\mathbf{t}_i)}{\|\mathcal{I}(\mathbf{x})\| \cdot \|\mathcal{T}(\mathbf{t}_i)\|}. \tag{3}$$

where $\mathbf{t}_i = \text{prompt} < y_i >$ and $y_i \in \mathcal{Y}^{\text{id}} \cup \mathcal{Y}^{\text{ood}}$, and $\mathbf{t}_i$ represents the textual description of the label $y_i$, using a prompt format such as "A photo of a <label>."

**Fusion of Similarity Scores.** To fully utilize both image-to-image and image-to-text similarity information, we compute a fused similarity score using a weighted sum-softmax method:

$$S_{\text{DualCnst}}(\mathbf{x}) = \sum_{i=1}^{K} \frac{\exp(\tilde{s}_i(\mathbf{x}))}{\sum_{j=1}^{K+M} \exp(\tilde{s}_j(\mathbf{x}))}, \tag{4}$$

where the fused similarity score $\tilde{s}_i(\mathbf{x})$ is defined as:

$$\tilde{s}_i(\mathbf{x}) = \alpha \cdot s_{i,\text{img}}(\mathbf{x}) + (1 - \alpha) \cdot s_{i,\text{text}}(\mathbf{x}), \tag{5}$$

where $\alpha$ is a fusion hyperparameter that balances the contributions of image-to-image and image-to-text similarities. Details on the choice of $\alpha$ are provided in Appendix G.6.

**OOD Detection Framework.** Based on $S_{\text{DualCnst}}(\mathbf{x})$, the OOD detector $G_\lambda(\mathbf{x}; \mathcal{Y}^{\text{id}} \cup \mathcal{Y}^{\text{ood}}, \mathcal{T}, \mathcal{I})$ is defined as a binary classification function:

$$G_\lambda(\mathbf{x}; \mathcal{Y}^{\text{id}} \cup \mathcal{Y}^{\text{ood}}, \bar{\mathcal{X}}, \mathcal{T}, \mathcal{I}) = \begin{cases} \text{ID} & S_{\text{DualCnst}}(\mathbf{x}) \geq \lambda \\ \text{OOD} & S_{\text{DualCnst}}(\mathbf{x}) < \lambda \end{cases}, \tag{6}$$

where $\lambda$ is a threshold selected such that a high fraction of ID samples (typically 95%) exceed this value. See Algorithm 1 for the complete zero-shot OOD detection procedure.

# 4 Theoretical Analysis of Multimodal Negative Label Aggregation for OOD Separability

We provide a simplified probabilistic analysis to illustrate how incorporating multimodal negative labels reduces the variance of similarity scores and improves ID–OOD separability. The key idea is to enrich the negative label space by augmenting existing textual labels with corresponding synthesized image labels across multiple feature levels. To better understand how such multimodal expansion improves OOD detection, we frame the problem from the perspective of multi-label classification and derive a theoretical formulation of detection performance.

Let the multimodal negative label set be defined as $\widetilde{Y}_i = \{\widetilde{y}_{i,1}, \widetilde{y}_{i,2}, \ldots, \widetilde{y}_{i,N}\} \in \mathcal{Y}^{\text{ood}}$, where $\widetilde{y}_{i,1}$ denotes the primary-modality negative label (e.g., text), and $\widetilde{y}_{i,2}, \ldots, \widetilde{y}_{i,N}$ represent auxiliary-modality labels (e.g., synthetic images encoded through image encoder $\mathcal{I}$ at different layers: $\widetilde{y}_{i,j} = \mathcal{I}^{(j-1)}(\bar{\mathbf{x}}_i)$). We compute the similarity between the test image $\mathbf{x}$ and each negative label $y_{i,j}$, denoted by $s_{i,j}$. Specifically, $s_{i,1} = s_{i,\text{text}}(\mathbf{x})$ corresponds to the text-based similarity, and $s_{i,j} = s_{i,\text{img}}^{(j-1)}(\mathbf{x})$ for $j > 1$ corresponds to image-based similarities (see Eq. (1)).

To aggregate these scores across modalities, a non-uniform weighting scheme is applied as defined in Eqs. (2) and (5). Without loss of generality, we assign a fraction $a/N$ (where $N > a > 1$) to the primary modality and distribute the remaining weight evenly among the auxiliary modalities:

$$
w_j = \begin{cases} \frac{a}{N}, & j = 1 \quad \text{(primary modality weight)} \\ \frac{1 - \frac{a}{N}}{N-1}, & j = 2, \ldots, N \quad \text{(auxiliary modality weights)} \end{cases}
$$

The aggregated similarity score $s_i$ is then computed as a weighted sum:

$$
s_i = \frac{a}{N} s_{i,1} + \sum_{j=2}^{N} \left( \frac{1 - \frac{a}{N}}{N-1} \right) s_{i,j}.
$$

Assuming that similarity scores $s_{i,j}$ are independently and identically distributed (i.i.d.) with expectation $\mathbb{E}[s_{i,j}] = \mu$ and variance $\text{Var}(s_{i,j}) = \sigma^2$, we derive:

$$
\mathbb{E}[s_i] = \mu, \quad \text{Var}(s_i) = \frac{\sigma^2}{N'(N)}, \quad \text{where} \quad N'(N) := \frac{N(N-1)}{a^2 + N - 2a}.
$$

By the Central Limit Theorem (CLT), for sufficiently large $N$, the distribution of $s_i$ can be approximated by

$$
s_i \sim \mathcal{N}\left( \mu, \frac{\sigma^2}{N'(N)} \right).
$$

This results in the probability expression:

$$
p(N') = P(s_i \geq \psi) = 1 - \Phi\left( \frac{\psi - \mu}{\sigma} \sqrt{N'(N)} \right),
$$

where $\Phi(\cdot)$ denotes the cumulative distribution function (CDF) of the standard normal distribution. The binary indicator $s_i^*$ is then defined as $s_i^* = \mathbb{I}[s_i \geq \psi]$ which serves as the basis for the OOD detection score. The negative label match count is then defined as $c = \sum_{i=1}^{M} s_i^*$, representing the OOD score, which follows a binomial distribution $B(M, p(N'))$. Specifically, the match counts for ID and OOD samples are given by $c_{\text{in}} \sim B(M, p_1(N'))$ and $c_{\text{out}} \sim B(M, p_2(N'))$, where $p_1(N')$ and $p_2(N')$ represent the distributions of the aggregated similarity scores $s_{i,j}$ for ID and OOD samples, respectively:

$$
p_1(N') = 1 - \Phi\left( \frac{\psi - \mu_1}{\sigma} \sqrt{N'} \right), \quad p_2(N') = 1 - \Phi\left( \frac{\psi - \mu_2}{\sigma} \sqrt{N'} \right)
$$

where $(\mu_1, \sigma)$ and $(\mu_2, \sigma)$ denote the mean and standard deviation of similarity scores for ID and OOD, respectively. Since OOD samples typically exhibit higher similarity with OOD labels than ID samples, it follows that $\mu_2 > \mu_1$. Consequently, the fundamental inequality $p_2 > p_1$ holds due to the monotonicity of the standard normal CDF $\Phi(\cdot)$.

For sufficiently large $M$, the CLT allows for the normal approximations

$$c_{\text{in}} \sim \mathcal{N}(Mp_1, Mp_1(1-p_1)), \quad c_{\text{out}} \sim \mathcal{N}(Mp_2, Mp_2(1-p_2)).$$

To evaluate the separability between ID and OOD samples, the false positive rate $\text{FPR}_\lambda$ is introduced as a performance metric of the OOD detector:

$$\text{FPR}_\lambda = F_{\text{out}}\left(F_{\text{in}}^{-1}(\lambda)\right),$$

where $F_{\text{in}}$ and $F_{\text{out}}$ are the CDFs of $c_{\text{in}}$ and $c_{\text{out}}$, respectively, and $\lambda \in [0,1]$ represents the true positive rate (TPR), quantifying the proportion of correctly classified ID samples. This yields the expression:

$$\text{FPR}_\lambda = \Phi\left(\frac{Mp_1(N') + \sqrt{Mp_1(N')(1-p_1(N'))} \cdot \Phi^{-1}(\lambda) - Mp_2(N')}{\sqrt{Mp_2(N')(1-p_2(N'))}}\right).$$

To examine the relationship between $\text{FPR}_\lambda$ and the number of multimodal labels $N$, the partial derivative $\partial \text{FPR}_\lambda / \partial N'$ is computed. Assuming $\mu_2 > \mu_1$ and $\mu_1 + \sigma/\sqrt{N'} > \mu_2$, the result shows that $\partial \text{FPR}_\lambda / \partial N' < 0$ when $\mu_1 + \sigma/\sqrt{N'u} > \psi > \mu_2$. Since $N'$ increases with $N$, it follows that $\text{FPR}_\lambda$ decreases with $N$, demonstrating that incorporating more auxiliary modalities in negative labels enhances OOD detection performance.

# 5 Experiments

## 5.1 Experiment Setup

**Datasets and Benchmarks.** For our experiments, we use ImageNet-1k [30] as the primary ID dataset. OOD datasets include iNaturalist [11], SUN [31], Places [32], and Textures [33], which cover a wide variety of scenes and semantic categories. We also adopt the experimental setup from MCM [9], which leverages subsets of ImageNet-1k to evaluate our method. Specifically, ImageNet-10 and ImageNet-20 are alternately used as ID and OOD datasets. Furthermore, we extend our evaluation to more generalized ImageNet variants, including ImageNet-R [34].

**Implementation Details.** Our framework is built upon CLIP [8] as the core model. Unless otherwise noted, we utilize the ViT-B/16 architecture as the image encoder and a Masked Self-Attention Transformer [35] as the text encoder. For image generation, we employ the Stable Diffusion. We set $\alpha = 0.2$ and $w = 0.15$, and provide ablation experiments. Further details can be found in Appendix G. To improve inference efficiency, all synthetic images are pre-generated before the evaluation phase, eliminating the need for additional computational overhead during testing. Further details in Appendix F.3.

For evaluation, we use two primary metrics: (1) **FPR95**: The false positive rate (FPR) at a true positive rate (TPR) of 95% for ID data. (2) **AUROC**: The area under the receiver operating characteristic curve. Additionally, we report the results in terms of **AUPR** in Appendix F.2.

**Baseline Methods.** We benchmark our method against several state-of-the-art zero-shot OOD detection approaches, including Mahalanobis Distance [36], Energy Score [37], ZOC [38], MCM [9], and NegLabel [12]. Additionally, we compare our approach with OOD detection models that have been trained or fine-tuned using ID data, such as MOS [39], MSP [1], CLIPN [40], VOS [41], and NPOS [42].

## 5.2 Main Results

**Far-OOD Detection on ImageNet-1k.** We compare our method with representative OOD detection approaches—including zero-shot methods (MCM, EOE, NegLabel) and supervised re-implementations that fine-tune CLIP on ImageNet-1k—summarized in Table 1. Our approach attains the strongest overall performance on ImageNet-1k: relative to the strongest prior method, NegLabel, it reduces average FPR95 by 1.75% and increases average AUROC by 0.14%, and it outperforms NegLabel on *all* evaluated OOD datasets. Robustness under domain shifts is analyzed in Appendix E.1.

**Near-OOD Detection on ImageNet Subsets.** We further report near-OOD results on the ImageNet-10/ImageNet-20 subsets in the main text (Table 2). When ImageNet-10 is treated as the in-distribution

Table 1: Performance Comparison of ImageNet-1k on Far OOD Detection. The **bold** indicates the best performance on each dataset, and the gray indicates methods requiring an additional massive auxiliary dataset.

| Method | iNaturalist FPR95↓ | iNaturalist AUROC↑ | SUN FPR95↓ | SUN AUROC↑ | Places FPR95↓ | Places AUROC↑ | Textures FPR95↓ | Textures AUROC↑ | Average FPR95↓ | Average AUROC↑ |
|---|---|---|---|---|---|---|---|---|---|---|
| MOS (BiT) [39] | 9.28 | 98.15 | 40.63 | 92.01 | 49.54 | 89.06 | 60.43 | 81.23 | 39.97 | 90.11 |
| MSP [1] | 40.89 | 88.63 | 65.81 | 81.14 | 67.90 | 80.14 | 64.96 | 78.16 | 59.89 | 82.04 |
| CLIPN [40] | 19.13 | 96.20 | 25.69 | 94.18 | 32.14 | **92.26** | 44.60 | 88.93 | 30.39 | 92.89 |
| VOS [41] | 28.99 | 94.62 | 36.88 | 92.57 | 38.39 | 91.23 | 61.02 | 86.33 | 41.32 | 91.19 |
| NPOS [42] | 16.58 | 96.19 | 43.77 | 90.44 | 45.27 | 89.44 | 46.12 | 88.80 | 37.93 | 91.22 |
| Mahalanobis [36] | 99.33 | 55.89 | 99.41 | 59.94 | 98.54 | 65.96 | 98.46 | 64.23 | 98.94 | 61.50 |
| Energy [37] | 81.08 | 85.09 | 79.02 | 84.24 | 75.08 | 83.38 | 93.65 | 65.56 | 82.21 | 79.57 |
| ZOC [38] | 87.30 | 86.09 | 81.51 | 81.20 | 73.06 | 83.39 | 98.90 | 76.46 | 85.19 | 81.79 |
| MCM [9] | 30.91 | 94.61 | 37.59 | 92.57 | 44.69 | 89.77 | 57.77 | 86.11 | 42.74 | 90.77 |
| NegLabel [12] | 1.91 | 99.49 | 20.53 | 95.49 | 35.59 | 91.64 | 43.56 | 90.22 | 25.40 | 94.21 |
| DualCnst | **0.98** | **99.70** | **18.13** | **95.66** | **31.77** | 91.81 | **34.91** | **91.73** | **21.45** | **94.72** |

Table 2: Performance Comparison of ImageNet Subsets on Near OOD Detection. The **bold** indicates the best performance on each dataset, and the gray indicates methods requiring an additional massive auxiliary dataset.

| Method | ID OOD | ImageNet-10 ImageNet-20 FPR95↓ | ImageNet-10 ImageNet-20 AUROC↑ | ImageNet-20 ImageNet-10 FPR95↓ | ImageNet-20 ImageNet-10 AUROC↑ | Average FPR95↓ | Average AUROC↑ |
|---|---|---|---|---|---|---|---|
| CLIPN [40] | | 7.80 | 98.07 | 13.67 | 97.47 | 10.74 | 97.77 |
| MaxLogit [1] | | 9.70 | 98.09 | 14.00 | 97.81 | 11.85 | 97.95 |
| Energy [37] | | 10.30 | 97.94 | 16.40 | 97.37 | 13.35 | 97.66 |
| MCM [9] | | 5.00 | 98.71 | 17.40 | **97.87** | 11.20 | **98.29** |
| NegLabel [12] | | 5.10 | 98.86 | 17.60 | 97.04 | 11.35 | 97.95 |
| DualCnst | | **2.20** | **98.96** | **12.20** | 97.44 | **7.45** | 98.20 |

(ID) dataset and ImageNet-20 as OOD, our method lowers FPR95 by 2.4% and improves AUROC by 0.1% over NegLabel. Conversely, when ImageNet-20 is ID and ImageNet-10 is OOD, our method achieves a 5.4% reduction in FPR95 and a 0.4% gain in AUROC. The subset splits and ID label configurations follow MCM [9]. For a fair comparison, we reproduce NegLabel and MCM under the same protocol.

**Far-OOD Across Alternative ID Datasets** To assess generality beyond ImageNet-1k and to understand how ID semantics affect far-OOD detection, we evaluate our method across seven ID datasets with diverse granularity and domain bias: *CUB-200-2011* (fine-grained birds) [43], *Stanford-Cars* (fine-grained vehicles) [44], *Food-101* (coarse-to-fine foods) [45], *Oxford-IIIT Pet* (fine-grained pets) [46], and three ImageNet-derived subsets (*ImageNet-10*, *ImageNet-20*, *ImageNet-100*). For each ID dataset, we evaluate on the standard far-OOD suite *iNaturalist*, *SUN*, *Places*, and *Texture*, reporting FPR95 ↓ and AUROC ↑ (Table 3). Unless otherwise specified, we use the same CLIP backbone as prior work and do not tune the backbone on OOD data. We set $\alpha = 0.1$, extract features from the 3rd, 6th, and 9th layers of the visual encoder, and use a fusion weight of $w = 0.15$. All hyperparameters are selected *once* using ID-only validation and are shared across datasets to avoid per-dataset cherry picking.

Across all seven ID datasets and four OOD benchmarks, our method surpasses the strongest baseline. On fine-grained IDs such as *Stanford-Cars* and *Oxford-IIIT Pet*, performance approaches the numerical ceiling (near-zero FPR95 with AUROC $\approx$ 100), indicating that our dual-constraint design preserves class discrimination while suppressing spurious activations. On more heterogeneous IDs (e.g., *ImageNet-100*), our approach yields lower false-positive rates than NegLabel while maintaining high AUROC, suggesting improved calibration under broader intra-class variation. Notably, gains are most pronounced on *Texture* and *Places*, which are known to challenge methods that overly rely on background or style cues.

### 5.3 Ablation Study

**Score Functions.** To demonstrate the superiority of the proposed OOD detection score $S_{\text{DualCnst}}$, we present the average results on the ImageNet-1K dataset in Figure 3 (a), comparing it with other

Table 3: Performance Comparison of Different ID Datasets on Far OOD Detection. The **bold** indicates the best performance on each dataset.

| ID Dataset | Method | iNaturalist | | SUN | | Places | | Texture | | Average | |
|---|---|---|---|---|---|---|---|---|---|---|---|
| | | FPR95↓ | AUROC↑ | FPR95↓ | AUROC↑ | FPR95↓ | AUROC↑ | FPR95↓ | AUROC↑ | FPR95↓ | AUROC↑ |
| Stanford-Cars | MCM [9] | 0.05 | 99.77 | 0.02 | 99.95 | 0.24 | 99.89 | 0.02 | 99.96 | 0.08 | 99.89 |
| | NegLabel [12] | 0.01 | 99.99 | 0.01 | 99.99 | 0.03 | 99.99 | 0.01 | 99.99 | 0.01 | 99.99 |
| | DualCnst | **0.00** | **100.00** | **0.00** | **100.00** | **0.03** | **99.99** | **0.00** | **100.00** | **0.01** | **100.00** |
| CUB-200 | MCM [9] | 9.83 | 98.24 | 4.93 | 99.10 | 6.65 | 98.57 | 6.97 | 98.75 | 7.09 | 98.66 |
| | NegLabel [12] | 0.18 | 99.96 | 0.02 | 99.99 | 0.33 | **99.90** | 0.01 | 99.99 | 0.13 | 99.96 |
| | DualCnst | **0.12** | **99.98** | **0.02** | **99.99** | 0.38 | 99.89 | **0.00** | **100.00** | **0.13** | 99.96 |
| Oxford-Pet | MCM [9] | 2.85 | 99.38 | 1.06 | 99.73 | 2.11 | 99.56 | 0.80 | 99.81 | 1.70 | 99.62 |
| | NegLabel [12] | 0.01 | 99.99 | 0.02 | 99.99 | 0.17 | 99.96 | 0.11 | 99.97 | 0.07 | 99.98 |
| | DualCnst | **0.00** | **100.00** | **0.00** | **100.00** | **0.15** | **99.97** | **0.09** | **99.98** | **0.06** | **99.99** |
| Food-101 | MCM [9] | 0.64 | 99.78 | 0.90 | 99.75 | 1.86 | 99.58 | 4.04 | 98.62 | 1.86 | 99.43 |
| | NegLabel [12] | 0.01 | 99.99 | 0.01 | 99.99 | 0.01 | 99.99 | 1.61 | **99.60** | 0.40 | **99.90** |
| | DualCnst | **0.00** | **100.00** | **0.00** | **100.00** | **0.01** | **100.00** | **1.52** | 99.57 | **0.38** | 99.89 |
| ImageNet-10 | MCM [9] | 0.12 | 99.80 | 0.29 | 99.79 | 0.88 | 99.62 | 0.04 | 99.90 | 0.33 | 99.78 |
| | NegLabel [12] | 0.02 | 99.83 | 0.20 | 99.88 | 0.71 | 99.75 | 0.02 | 99.94 | 0.24 | 99.85 |
| | DualCnst | **0.01** | **99.97** | **0.09** | **99.93** | **0.57** | **99.75** | **0.02** | **99.96** | **0.17** | **99.90** |
| ImageNet-20 | MCM [9] | 1.02 | 99.66 | 2.55 | 99.50 | 4.40 | 99.11 | 2.43 | 99.03 | 2.60 | 99.32 |
| | NegLabel [12] | 0.15 | 99.95 | 1.93 | 99.51 | 4.40 | 98.97 | 2.41 | 99.11 | 2.22 | 99.39 |
| | DualCnst | **0.13** | **99.97** | **1.22** | **99.66** | **3.66** | **99.13** | **2.18** | **99.17** | **1.80** | **99.48** |
| ImageNet-100 | MCM [9] | 18.13 | 96.77 | 36.45 | 94.54 | 34.52 | 94.36 | 41.22 | 92.25 | 32.58 | 94.48 |
| | NegLabel [12] | 0.53 | 99.87 | 9.91 | 98.12 | 20.26 | 96.18 | 25.50 | 95.27 | 14.05 | 97.36 |
| | DualCnst | **0.41** | **99.90** | **8.68** | **98.34** | **18.72** | **96.43** | **23.51** | **95.72** | **12.83** | **97.60** |

scoring functions: $S_{MAX}$, $S_{Energy}$, and $S_{MaxLogit}$. All these functions are specifically designed for the Dual Consistency approach. Please refer to Appendix G.12 for the specific forms and results on more datasets. Results show that our $S_{DualCnst}$ achieves the best OOD performance. This verifies the superiority and importance of the proposed OOD detection score.

**Different Layers of the Visual Encoder.** To explore the effectiveness of pixel-level features from different layers of the visual encoder, we sample various pixel layers and assign different weights, as shown in Figure 3 (b). Specifically, we experiment by selecting the (1st, 2nd, 3rd) layers, (4th, 5th, 6th) layers, (7th, 8th, 9th) layers, (9th, 10th, 11th) layers, and all pixel layers to combine with semantic layers. In Figure 3 (c), we further investigate the impact of different weight distributions for $w$ to identify the most suitable pixel-level feature weighting. For details on the selection of $w$, layers, and results, refer to Appendix G.5.

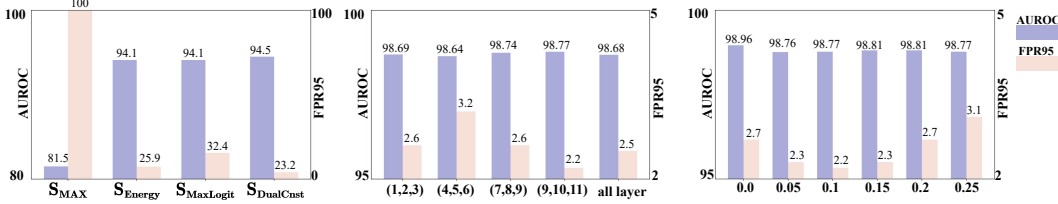

Figure 3: Ablation study on (a) score function, (b) Different Layers, and (c) Different Weight. ID dataset: ImageNet-10; OOD dataset: ImageNet-20.

## 5.4 Further Analysis

**More Experimental Results**. We conducted experiments on the CIFAR-10/CIFAR-100 [10] benchmark to further validate our method. The details of the ImageNet-A [47] and ImageNet-V2 [48] generalization datasets are also provided in the Appendix E.2. Additionally, we explored the impact of randomness introduced by Stable Diffusion when generating synthetic images with different random seeds, as demonstrated in Table 33 . The results show that the effect of Stable Diffusion's randomness on our method is negligible. It is important to note that we did not manually select the most favorable random seed for Stable Diffusion. Instead, we generated a 32-bit integer random seed by hashing the combination of each class label and synthetic image index. Each synthetic image generated for a class using this seed exhibits substantial randomness, further demonstrating that our method is not

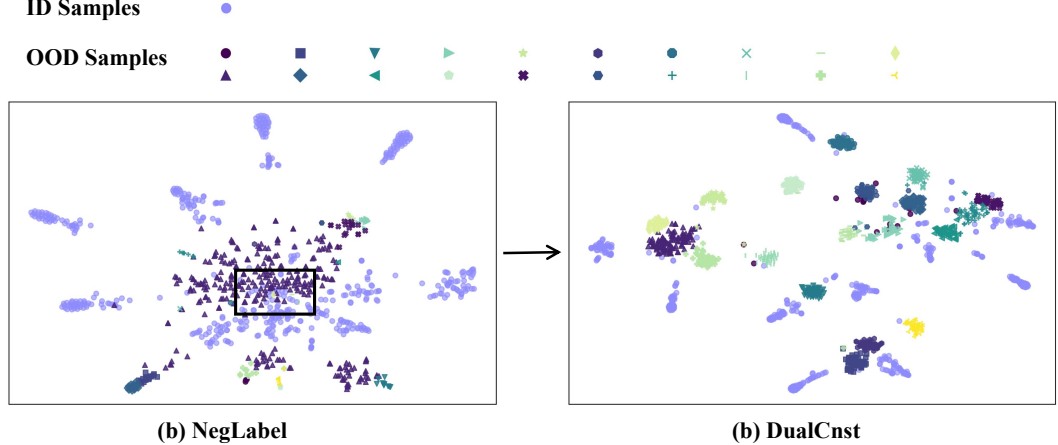

**(b) NegLabel**     **(b) DualCnst**

Figure 4: T-SNE visualizations obtained by the classifier output. ID set: ImageNet-10; OOD set: ImageNet-20. We use distinct colors to represent different OOD classes. Our DualCnst method achieves better separability between ID and OOD classes compared to NegLabel.

influenced by the randomness of Stable Diffusion-generated images. We also conducted experiments with different CLIP visual encoders, and the results showed that stronger visual encoders, which capture more detailed information, are more beneficial to our method. For more details, please refer to Appendix G.3.

**Effectiveness of DualCnst.** Figure 4 shows the T-SNE [49] visualization of the softmax outputs. We compare the results of NegLabel and DualCnst, using the ImageNet-10 dataset for ID and ImageNet-20 dataset for OOD. In this setup, there are several semantically similar pairs of ID and OOD categories, such as: horse (ID) vs. zebra (OOD), Swiss mountain dog (ID) vs. timberwolf (OOD), warplane (ID) vs. space shuttle (OOD), and garbage truck (ID) vs. steam locomotive (OOD). In the presence of such datasets, methods that expand the label space, like NegLabel, often struggle to find labels with a high overlap probability with true OOD labels, leading to suboptimal performance. As shown in (a) with the black bounding box, it is difficult to distinguish between ID and OOD samples, as they tend to interweave. DualCnst, however, addresses this issue by leveraging visual information to differentiate between ID and OOD samples. As demonstrated in (b), we incorporate visual information into NegLabel, allowing for better differentiation based on unique visual features inherent to ID and OOD samples, such as the stripes on a zebra or the ears and fur of a timberwolf. These observations indicate that DualCnst enables a significant improvement in the classifier's ability, making semantically similar ID and OOD samples more separable.

## 6  Conclusion

This work presented DualCnst, a training-free framework for zero-shot OOD detection that integrated both semantic and visual consistency. By generating synthetic exemplars from ID and OOD labels, it enabled robust similarity computation without requiring access to ID data or additional training. A unified scoring function fused semantic and visual cues, and theoretical analysis demonstrated that multimodal negative label aggregation reduced score variance and enhanced ID–OOD separability. Extensive experiments across diverse benchmarks confirmed that DualCnst outperformed existing zero-shot approaches and achieved state-of-the-art performance. By unifying vision-language alignment with generative visual synthesis in a training-free manner, DualCnst offered a scalable, plug-and-play solution for OOD detection in open-world scenarios.

## Acknowledgements

This work is in part supported by Fujian Province Natural Science Foundation of China under Grant (2024J01158) and Shenzhen Science and Technology Program under grant (JCYJ20240813155840052), and the National Key R&D Program of China (2022YFA1004203, 2021YFF0501503), the China's National Foundation for Natural Sciences (62125111, 62331028, 62476268, 62206273).

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

# Appendix

# A Broader Impacts

**Positive Impacts:** Our framework could enhance the reliability of AI systems in safety-critical domains like medical imaging, where undetected OOD samples may lead to diagnostic errors. By enabling synthetic data-driven OOD detection without real patient data, it also mitigates privacy risks associated with medical dataset sharing.

**Negative Impacts:** Malicious actors could exploit our text-to-image synthesis pipeline to generate adversarial OOD samples that evade detection. Additionally, energy-intensive image generation processes may contribute to environmental costs if scaled without optimization.

**Mitigations:** We recommend (1) adversarial robustness testing against synthetic OOD benchmarks, (2) controlled access to synthetic image generators in high-stakes applications.

# B Related Works

**OOD Detection.** Early methods for OOD detection include classification-based approaches that rely on a well-trained ID classifier, such as MSP [1]. Density-based methods, such as likelihood ratios [50] and likelihood regret [51], estimate the likelihood of data points to identify OOD samples. Reconstruction-based methods [52, 53, 54] leverage reconstruction errors from generative models, including VAEs and autoencoders, to detect OOD instances. Post-hoc methods, including ODIN [55] and energy-based scoring [37], enhance pre-trained models without modifying their parameters. More recently, multimodal vision-language models such as CLIP and its variants [56] have enabled zero-shot OOD detection by leveraging text-image embeddings, marking a shift toward more versatile and scalable solutions.

**Zero-shot OOD Detection.** Recent advancements in zero-shot OOD detection take advantage of the powerful pretraining capabilities of models like CLIP, allowing for efficient OOD detection without the need for large external OOD labels. ZOC [38] introduces a CLIP-based framework for zero-shot OOD detection, where potential OOD labels are generated for input instances using image captions, aligning images and text for zero-shot classification. MCM [9] performs OOD detection by utilizing scaled softmax values of the maximum logits as confidence scores, but it relies solely on ID class labels and does not fully exploit open-world textual information. CLIPN [40] improves the model's ability to reject mismatched inputs by introducing learnable "negative" prompts and a dedicated "negative" text encoder. EOE [57] utilizes the expert knowledge and reasoning abilities of large language models (LLMs) to generate potential anomalies, enabling more effective OOD detection. NegLabel [12] proposes a novel method that enhances the distinguishability between ID and OOD samples by mining potential OOD labels from a corpus. However, these methods do not fully consider the visual effectiveness of images. In contrast, DualCnst addresses this limitation by making semantically similar ID and OOD samples more distinguishable. Moreover, it can be seamlessly integrated into existing OOD frameworks.

**Stable Diffusion for OOD Detection.** Stable Diffusion has been explored for OOD detection in several studies. [58] introduced a semantic mismatch-guided variant by masking input regions and measuring semantic inconsistencies between original and reconstructed samples, which addresses the limitation of pixel-level error metrics in traditional reconstruction methods. Diffusion-based neighborhood analysis [59] injects noise perturbations to generate sample variants and quantifies latent feature distribution divergence for enhanced sensitivity to local anomalies. LMD [54] introduces a diffusion-based approach for image inpainting, where the input image is reconstructed, and the reconstruction error is used as an indicator for OOD detection. In contrast, DualCnst employs Stable Diffusion for image generation, offering a more efficient solution in open-world scenarios. Unlike LMD, DualCnst reduces the computational burden on the inference process, making it a more practical and scalable approach for OOD detection in dynamic environments.

**Systematic Comparative Analysis.** To better characterize methodological distinctions, we conduct a systematic comparison across three key dimensions: (1) Enhancement strategy spectrum (semantic/image space), (2) Training requirement compatibility, and (3) Multimodal-generative integration. As Table 4 demonstrates, existing approaches exhibit notable limitations: classification baselines lack multimodal awareness, zero-shot methods neglect visual space enhancement, while generative approaches impose substantial training overhead. DualCnst uniquely combines semantic-textual align-

---

**Algorithm 1** Zero-shot OOD detection with text-image dual consistency

---

1: **Input:** ID class labels $\mathcal{Y}^{\text{id}}$, test sample $\mathbf{x}$, text encoder $\mathcal{T}$, image encoder $\mathcal{I}$, Stable Diffusion (SD), NegLabel, fusion coefficient $\alpha$, layer weight $w$, threshold $\lambda$;
   **Synthesis stage:**
   // Synthesize OOD class labels
2: Given $\mathcal{Y}^{\text{id}}$, $\mathcal{Y}^{\text{ood}} = \text{NegLabel}(\mathcal{Y}^{\text{id}})$;
   // Synthesize ID/OOD image labels
3: Given $\mathcal{Y}^{\text{id}} \cup \mathcal{Y}^{\text{ood}}$, $\bar{\mathcal{X}} = \text{SD}(\text{prompt} < \mathcal{Y}^{\text{id}} \cup \mathcal{Y}^{\text{ood}} >)$;
   **Testing stage:**
   // Calculate image-to-image similarity
4: $s_{i,\text{img}}^{(l)}(\mathbf{x}) = \frac{\mathcal{I}^{(l)}(\mathbf{x}) \cdot \mathcal{I}^{(l)}(\bar{\mathbf{x}}_i)}{\|\mathcal{I}^{(l)}(\mathbf{x})\| \cdot \|\mathcal{I}^{(l)}(\bar{\mathbf{x}}_i)\|}$; $\bar{\mathbf{x}}_i \in \bar{\mathcal{X}}$;
5: $s_{i,\text{img}}(\mathbf{x}) = \sum_{l=1}^{L} w_l \cdot s_{i,\text{img}}^{(l)}(\mathbf{x})$;
   // Calculate image-to-text similarity
6: $\mathbf{t}_i = \text{prompt}<y_i>$; $y_i \in \mathcal{Y}^{\text{id}} \cup \mathcal{Y}^{\text{ood}}$;
7: $s_{i,\text{text}}(\mathbf{x}) = \frac{\mathcal{I}(\mathbf{x}) \cdot \mathcal{T}(\mathbf{t}_i)}{\|\mathcal{I}(\mathbf{x})\| \cdot \|\mathcal{T}(\mathbf{t}_i)\|}$;
   // Integrate text and visual information
8: $\tilde{s}_i(\mathbf{x}) = \alpha \cdot s_{i,\text{img}}(\mathbf{x}) + (1 - \alpha) \cdot s_{i,\text{text}}(\mathbf{x})$;
   // Calculate OOD detection score
9: $S_{\text{DualCnst}}(\mathbf{x}) = \sum_{i=1}^{K} \frac{\exp(\tilde{s}_i(\mathbf{x}))}{\sum_{j=1}^{K+M} \exp(\tilde{s}_j(\mathbf{x}))}$;
10: **Output: ID** if $S_{\text{DualCnst}}(\mathbf{x}) > \lambda$, else **OOD**.

---

ment with visual-space consistency learning, achieving full compliance with all evaluation criteria–the only method supporting both training-free deployment and multimodal-generative synergy.

Table 4: Comparative Analysis of Method Characteristics

| Method Category | Representative Methods | Enhancement Strategy | | Training-Free | Multimodal Utilization | Generative Model Application |
| --- | --- | --- | --- | --- | --- | --- |
| | | Semantic Space | Image Space | | | |
| Classification Benchmarks | MSP [1] | × | × | × | × | × |
| | ODIN [55] | × | × | ✓ | × | × |
| | Energy [37] | × | × | ✓ | × | × |
| Zero-Shot Methods | ZOC [38] | ✓ | × | ✓ | ✓ | × |
| | MCM [9] | × | × | ✓ | × | × |
| | CLIPN [40] | ✓ | × | ✓ | ✓ | × |
| | NegLabel [12] | ✓ | × | ✓ | ✓ | × |
| Generative Approaches | Yang [58] | × | ✓ | × | × | ✓ |
| | LMD [54] | × | ✓ | × | × | ✓ |
| | Diffusion [59] | × | ✓ | ✓ | × | ✓ |
| **DualCnst (Ours)** | | ✓ | ✓ | ✓ | ✓ | ✓ |

Notation: ✓=Supported, ×=Not Supported. Text in parentheses indicates implementation specifics.

# C  Theoretical Justification for Section 4

## C.1  Fusion under weak dependence

To analyze the dependence of the false positive rate $\text{FPR}_\lambda$ on the number of multimodal labels $N$, consider its reformulation in terms of the error function $\text{erf}(\cdot)$:

$$\text{FPR}_\lambda = \frac{1}{2} + \frac{1}{2}\text{erf}\left(\sqrt{\frac{p_1(N')(1 - p_1(N'))}{p_2(N')(1 - p_2(N'))}} \cdot \text{erf}^{-1}(2\lambda - 1) + \frac{\sqrt{M}(p_1(N') - p_2(N'))}{\sqrt{2p_2(N')(1 - p_2(N'))}}\right),$$

where the error function is defined as $\mathrm{erf}(x) = \frac{2}{\sqrt{\pi}} \int_0^x e^{-t^2} dt$, which satisfies $\Phi(x) = \frac{1}{2}\left[1 + \mathrm{erf}\left(\frac{x}{\sqrt{2}}\right)\right]$. Define:

$$A = \sqrt{\frac{p_1(1-p_1)}{p_2(1-p_2)}}\mathrm{erf}^{-1}(2\lambda - 1), \quad B = \frac{\sqrt{M}(p_1 - p_2)}{\sqrt{2p_2(1-p_2)}},$$

where

$$p_1 = 1 - \Phi(k_1\sqrt{N'}), \quad p_2 = 1 - \Phi(k_2\sqrt{N'}),$$

and

$$k_1 = \frac{\psi - \mu_1}{\sigma}, \quad k_2 = \frac{\psi - \mu_2}{\sigma}.$$

Since $\mu_1 < \mu_2$ and $\psi > \mu_2$, it follows that $k_1 > k_2 > 0$. Now,

$$\frac{\partial A}{\partial N} = \mathrm{erf}^{-1}(2\lambda - 1) \cdot \frac{\partial}{\partial N}\sqrt{\frac{p_1(1-p_1)}{p_2(1-p_2)}}.$$

Letting $Q = \frac{p_1(1-p_1)}{p_2(1-p_2)}$, we obtain

$$\frac{\partial Q}{\partial N'} = \frac{(1-2p_1)\frac{\partial p_1}{\partial N'} \cdot p_2(1-p_2) - p_1(1-p_1)(1-2p_2)\frac{\partial p_2}{\partial N'}}{[p_2(1-p_2)]^2}.$$

From the definition of $p_1$ and $p_2$, the derivatives are given by

$$\frac{\partial p_1}{\partial N'} = -\phi(k_1\sqrt{N'}) \cdot \frac{k_1}{2\sqrt{N'}}, \quad \frac{\partial p_2}{\partial N'} = -\phi(k_2\sqrt{N'}) \cdot \frac{k_2}{2\sqrt{N'}},$$

where $\phi(x) = \frac{1}{\sqrt{2\pi}}e^{-x^2/2}$ is the standard normal probability density function (PDF). Since $\psi < \mu_1 + \frac{\sigma}{\sqrt{N'}}$, then $0 < k_2 < k_1 < \frac{1}{\sqrt{N'}}$, the function $\phi(k\sqrt{N'}) \cdot k$ is increasing with respect to $k$. Thus, we have

$$-\frac{\partial p_1}{\partial N'} > -\frac{\partial p_2}{\partial N'} > 0.$$

Moreover, since $\psi > \mu_2$ and $p_2 > p_1$, it follows that

$$p_2(1-p_2)(1-2p_1) > p_1(1-p_1)(1-2p_2) > 0.$$

Thus, we conclude that

$$\frac{\partial A}{\partial N'} < 0.$$

On the other hand,

$$\frac{\partial B}{\partial N'} = \frac{\sqrt{M}}{\sqrt{2}} \cdot \frac{\left(\frac{\partial p_1}{\partial N'} - \frac{\partial p_2}{\partial N'}\right)\sqrt{p_2(1-p_2)} - (p_1 - p_2)\cdot\frac{1-2p_2}{2\sqrt{p_2(1-p_2)}}\frac{\partial p_2}{\partial N'}}{p_2(1-p_2)}.$$

By the monotonicity of $\phi(k\sqrt{N'}) \cdot k$, it follows that

$$\left(\frac{\partial p_1}{\partial N'} - \frac{\partial p_2}{\partial N'}\right)\sqrt{p_2(1-p_2)} < 0.$$

Furthermore, since $p_2 > p_1$ and $p_2 < 1/2$, we obtain

$$-(p_1 - p_2) \cdot \frac{1-2p_2}{2\sqrt{p_2(1-p_2)}}\frac{\partial p_2}{\partial N} < 0.$$

Thus, it follows that

$$\frac{\partial B}{\partial N'} < 0.$$

By the definition of $\mathrm{FPR}_\lambda$,

$$\frac{\partial \mathrm{FPR}_\lambda}{\partial N'} = \frac{e^{-(A+B)^2}}{\sqrt{\pi}} \cdot \frac{\partial(A+B)}{\partial N'} < 0.$$

## C.2 Empirical Validation under Weak Dependence

We complement the theoretical analysis by empirically quantifying cross-modal and cross-layer dependencies and by examining how multimodal score fusion reduces variance. To this end, we compute pairwise correlations and covariances among similarity scores extracted from the text and image branches, as well as from multiple vision encoder layers. We additionally visualize these relationships in Figures 5 and 6 as heatmaps to illustrate the dependence structure across all layers.

**Cross-modal and cross-layer correlations.** Table 5 shows that the Pearson correlation between text-based and image-based similarities is nearly zero ($r = 0.01$), indicating that the two modalities convey largely independent information. Table 6 and the corresponding correlation heatmap in Figure 5 reveal that inter-layer dependencies are also weak: most off-diagonal correlations are close to zero and none exceed $0.2$ in magnitude. This observation empirically supports our weak-dependence assumption—different layers capture complementary but only mildly correlated cues.

Table 5: Pairwise Pearson correlations between modalities.

|  | img_sim | text_sim |
|---|---|---|
| img_sim | 1.00 | 0.01 |
| text_sim | 0.01 | 1.00 |

Table 6: Pairwise Pearson correlations across layers.

|  | text_sim | layer1_sim | layer5_sim | layer9_sim |
|---|---|---|---|---|
| text_sim | 1.00 | 0.00 | -0.12 | 0.04 |
| layer1_sim | 0.00 | 1.00 | 0.06 | -0.10 |
| layer5_sim | -0.12 | 0.06 | 1.00 | 0.19 |
| layer9_sim | 0.04 | -0.10 | 0.19 | 1.00 |

**Covariance matrices.** The covariance analysis in Tables 7–8 and Figure 6 further substantiates this independence trend. Cross-modal covariance is extremely small ($1.11 \times 10^{-5}$), indicating negligible shared variance between the text and image similarities. Across layers, off-diagonal covariances remain several orders of magnitude lower than diagonal terms (Table 8), implying that fluctuations between different similarity sources are largely uncorrelated.

The covariance heatmap in Figure 6 visually emphasizes this structure—bright diagonal dominance with near-zero off-diagonal intensity—demonstrating that each layer contributes distinct but weakly dependent information. This empirical finding supports the theoretical premise that aggregating such weakly dependent scores will monotonically reduce overall variance and yield a more stable OOD decision boundary.

Table 7: Covariance matrix between img_sim and text_sim.

|  | img_sim | text_sim |
|---|---|---|
| img_sim | 4.20e-04 | 1.11e-05 |
| text_sim | 1.11e-05 | 4.20e-04 |

Table 8: Covariance matrix across layers.

|  | text_sim | layer1_sim | layer5_sim | layer9_sim |
|---|---|---|---|---|
| text_sim | 4.20e-04 | -8.02e-07 | -1.88e-04 | 2.77e-06 |
| layer1_sim | -8.02e-07 | 6.08e-04 | 1.08e-04 | -9.57e-05 |
| layer5_sim | -1.88e-04 | 1.08e-04 | 5.10e-03 | 5.16e-04 |
| layer9_sim | 2.77e-06 | -9.57e-05 | 5.16e-04 | 1.50e-03 |

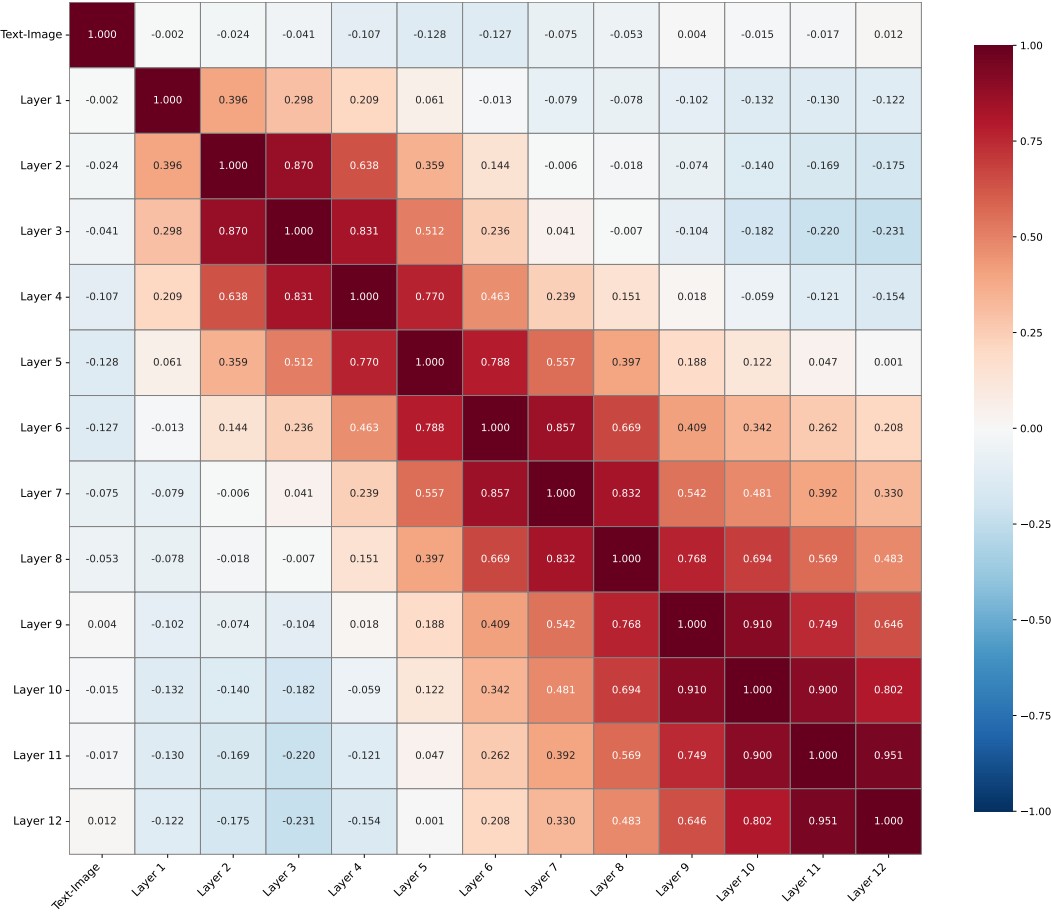

Figure 5: Heatmap of Pearson correlations across text and vision encoder layers. Off-diagonal elements remain close to zero, confirming weak cross-layer dependence.

**Variance reduction under fusion.** Finally, Table 9 quantifies the empirical variance of the aggregated score $S$ as more weakly dependent signals are fused. We observe a clear monotonic decline—from $0.711 \times 10^{-3}$ using only the text score to $0.257 \times 10^{-3}$ when six intermediate layers are integrated—demonstrating that fusing multiple partially independent cues effectively smooths stochastic fluctuations in similarity estimation. This empirical trend aligns with the theoretical prediction that $\mathrm{Var}(S) \propto 1/N$ under weak dependence, confirming that multimodal and multilayer fusion yields more stable and discriminative OOD scores.

Table 9: Empirical variance ($\times 10^{-3}$) vs. number of fused signals.

| Setting | Empirical variance |
|---|---|
| Only Text score | 0.7110 |
| Only Image score | 0.3691 |
| Fusion score (1 intermediate layer) | 0.3553 |
| Fusion score (2 intermediate layers) | 0.3366 |
| Fusion score (3 intermediate layers) | 0.3274 |
| Fusion score (4 intermediate layers) | 0.3038 |
| Fusion score (5 intermediate layers) | 0.2717 |
| Fusion score (6 intermediate layers) | 0.2567 |

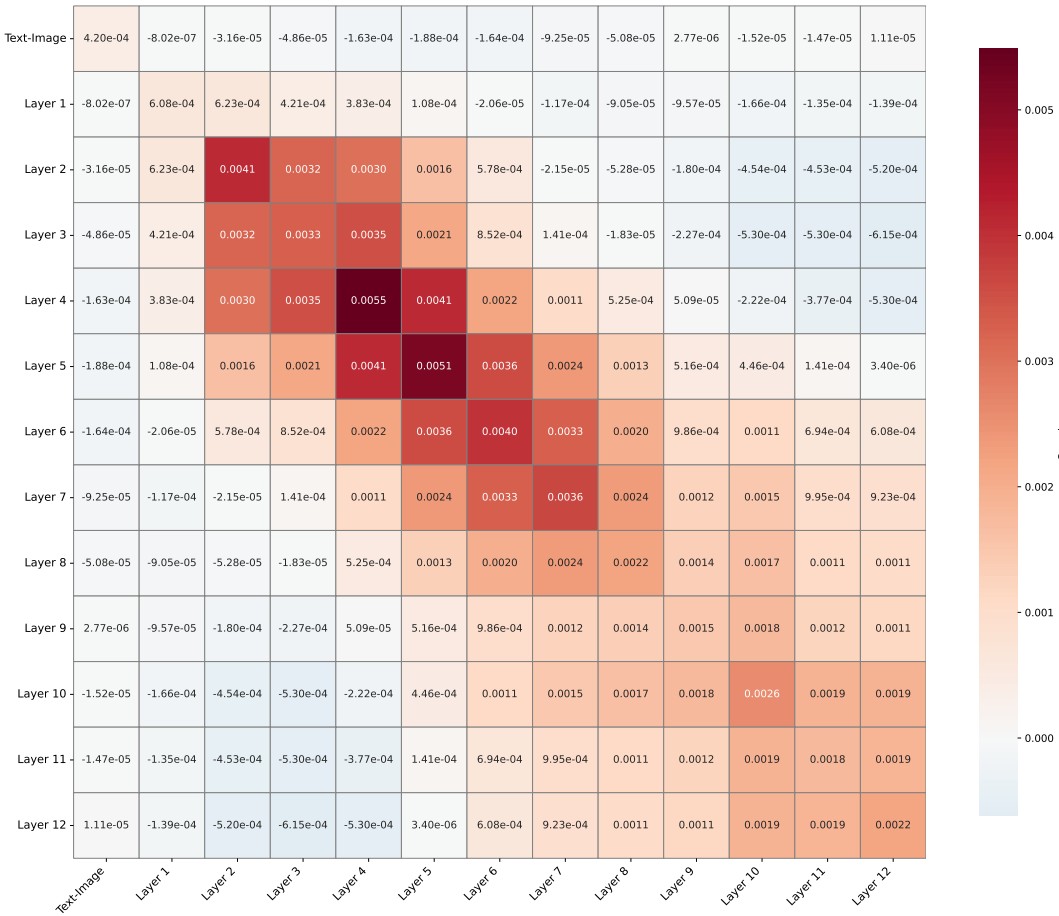

Figure 6: Heatmap of covariance values across text and vision encoder layers. Strong diagonal dominance and weak off-diagonal terms indicate low inter-layer variance coupling.

## D Limitations

**Limitation I. Dependence on synthetic image quality.** The proposed method relies on pre-generated synthetic images to capture visual exemplars for both in-distribution and OOD labels. Consequently, its performance may degrade if the generative model fails to produce high-quality images. In particular, artifacts such as severe noise, blur, or unstructured patterns may introduce misleading visual cues that hinder effective similarity computation. To address this, we recommend employing robust text-to-image models such as Stable Diffusion XL-Turbo, which substantially reduces generation failures. Notably, as shown in Table 31, even when the synthesized images are stylistically mismatched with the test samples (e.g., rendered as oil paintings instead of natural photos), they can still offer marginal performance gains over the text-only baseline.

**Limitation II. Computational cost of image synthesis.** The framework incurs additional computational overhead during the image synthesis stage. As reported in Table 18, generation time scales with the number of candidate OOD labels, which may pose challenges for large open-set label spaces. To mitigate this, we adopt the following optimizations: (1) using accelerated generative models such as Stable Diffusion XL-Turbo; and (2) pre-generating all synthetic exemplars offline before inference. Since image synthesis is performed once per label set and reused across test samples, the runtime cost during inference remains negligible, making the approach scalable to large datasets.

**Limitation III. Independence assumption (weak dependence in practice).** In Section 4, the theoretical analysis assumes i.i.d. similarity scores $s_{i,j}$ for clarity. This assumption can be relaxed: in practice, similarity signals across modalities and encoder layers are typically *weakly dependent*. Provided that the aggregated score satisfies a Lindeberg-type condition—or, more generally, arises

from a bounded-variance sequence with limited dependence (e.g., $m$-dependent or exchangeable variables)—a central-limit approximation remains valid. Crucially, the key insight persists: increasing the number $N$ of (appropriately weighted) modalities/layers reduces the variance of the fused score and improves separability; under weak dependence, the variance of the weighted sum continues to decay on the order of $1/N$, up to constants determined by the dependence structure. Empirically, we observe (i) small cross-modal and cross-layer Pearson correlations and (ii) a monotonic decrease in the empirical variance of the fused score as more signals are aggregated (Tables 5–9), supporting the practical validity of this relaxation.

# E  Further Experiments

## E.1  Robustness to Domain Shift

To assess the generalization ability of our method under domain shifts, we conducted experiments using the ImageNet Domain Shift dataset, with ImageNet-R serving as the ID dataset. Table 10 presents the results based on CLIP-B/16 with $\alpha = 0.1$, selecting the 3rd, 6th, and 9th layers of the visual encoder, and assigning a weight of $w = 0.15$. Our method demonstrates stronger generalization performance compared to NegLabel. ImageNet-R [34] consists of 30,000 images spanning 200 ImageNet categories, with representations in diverse artistic styles, including art, cartoons, graffiti, embroidery, graphics, origami, paintings, patterns, plastic objects, plush objects, sculptures, sketches, tattoos, toys, and video game renditions. Table 11 presents an evaluation of DualCnst's robustness using the ImageNet-A [47] generalization dataset as the ID dataset, while iNaturalist [11], SUN [31], Places [32], and Textures [33] serve as OOD datasets. We compare DualCnst against state-of-the-art methods. DualCnst outperforms NegLabel across all datasets, achieving an improvement of 2.09% in FPR95 and 0.25% in AUROC on average.

In Table 12, we further investigate the robustness of DualCnst under the same experimental setup using another generalization dataset, ImageNet-V2 [48]. The experimental results demonstrate that our proposed method exhibits superior performance in handling domain shifts.

Table 10: Robustness results on ImageNet-R dataset. The **black bold** indicates the best performance.

| Method | OOD Dataset | | | | | | | | Average | |
| | iNaturalist | | SUN | | Places | | Texture | | | |
| | FPR95↓ | AUROC↑ | FPR95↓ | AUROC↑ | FPR95↓ | AUROC↑ | FPR95↓ | AUROC↑ | FPR95↓ | AUROC↑ |
|---|---|---|---|---|---|---|---|---|---|---|
| Energy [37] | 99.91 | 30.36 | 99.33 | 33.20 | 98.84 | 34.74 | 99.56 | 23.09 | 99.41 | 30.35 |
| MaxLogit [1] | 86.53 | 81.58 | 82.11 | 81.48 | 78.16 | 79.86 | 91.24 | 69.45 | 84.51 | 78.09 |
| MCM [9] | 51.59 | 92.24 | 52.88 | 89.97 | 52.04 | 88.01 | 56.45 | 85.65 | 53.24 | 88.97 |
| NegLabel [12] | 1.60 | 99.58 | 15.77 | 96.03 | 29.48 | 91.97 | 35.67 | 90.60 | 20.63 | 94.54 |
| DualCnst | **0.59** | **99.86** | **8.92** | **98.19** | **19.27** | **95.20** | **14.13** | **95.50** | **10.73** | **97.19** |

Table 11: Robustness results on ImageNet-A dataset. The ID class labels are the same as ImageNet. The **black bold** indicates the best performance.

| Method | OOD Dataset | | | | | | | | Average | |
| | iNaturalist | | SUN | | Places | | Texture | | | |
| | FPR95↓ | AUROC↑ | FPR95↓ | AUROC↑ | FPR95↓ | AUROC↑ | FPR95↓ | AUROC↑ | FPR95↓ | AUROC↑ |
|---|---|---|---|---|---|---|---|---|---|---|
| Energy | 99.48 | 50.03 | 95.01 | 58.83 | 93.52 | 60.86 | 97.46 | 42.18 | 96.37 | 52.97 |
| MaxLogit | 92.88 | 74.14 | 81.54 | 80.55 | 78.51 | 79.06 | 90.00 | 69.41 | 85.73 | 75.79 |
| MCM | 80.41 | 77.02 | 76.12 | 78.92 | 76.90 | 76.48 | 74.10 | 77.36 | 76.88 | 77.45 |
| NegLabel | 4.09 | 98.80 | 44.38 | 89.83 | 60.10 | 82.88 | 64.34 | 80.25 | 43.23 | 87.94 |
| DualCnst | **3.54** | **98.99** | **32.41** | **92.79** | **48.66** | **87.04** | **47.77** | **89.54** | **33.09** | **92.09** |

Following the near-OOD evaluation protocol in OpenOOD v1.5, we conducted robustness evaluation using ImageNet-1k and ImageNet-R as in-distribution (ID) datasets, with SSB-Hard, NINCO, iNaturalist, and Texture as out-of-distribution (OOD) benchmarks. As summarized in Table 13, DualCnst demonstrates stronger robustness than baseline methods under this more challenging near-OOD experimental setup.

Table 12: Robustness results on ImageNet-V2 dataset. The ID class labels are the same as ImageNet. The **black bold** indicates the best performance.

| Method | iNaturalist | | SUN | | Places | | Texture | | Average | |
|---|---|---|---|---|---|---|---|---|---|---|
| | FPR95↓ | AUROC↑ | FPR95↓ | AUROC↑ | FPR95↓ | AUROC↑ | FPR95↓ | AUROC↑ | FPR95↓ | AUROC↑ |
| Energy | 99.85 | 32.93 | 99.12 | 34.45 | 98.02 | 39.51 | 99.57 | 21.52 | 99.14 | 32.10 |
| MaxLogit | 83.78 | 83.84 | 83.55 | 81.79 | 80.27 | 80.33 | 93.51 | 64.34 | 85.28 | 77.58 |
| MCM | 44.89 | 92.14 | 51.17 | 89.69 | 56.73 | 86.44 | 69.57 | 81.51 | 55.10 | 87.56 |
| NegLabel | 2.47 | 99.40 | 25.69 | 94.46 | 42.03 | 90.00 | **48.90** | **88.46** | 29.77 | 93.08 |
| DualCnst | **1.49** | **99.60** | **21.90** | **94.92** | **36.71** | **90.62** | 50.62 | 88.18 | **27.68** | **93.33** |

Table 13: Robustness experiments on ImageNet-1k and ImageNet-R. The **black bold** indicates the best performance.

| Method | SSB-Hard | | NINCO | | iNaturalist | | Texture | | Average | |
|---|---|---|---|---|---|---|---|---|---|---|
| | FPR95↓ | AUROC↑ | FPR95↓ | AUROC↑ | FPR95↓ | AUROC↑ | FPR95↓ | AUROC↑ | FPR95↓ | AUROC↑ |
| Energy [37] | 96.62 | 51.92 | 87.39 | 67.17 | 94.54 | 61.56 | 94.89 | 45.94 | 93.36 | 56.65 |
| MaxLogit [1] | 90.98 | 57.92 | 79.69 | 70.72 | 65.41 | 87.29 | 72.04 | 80.84 | 77.03 | 74.19 |
| MCM [9] | 94.00 | 57.42 | 88.98 | 63.40 | 75.39 | 83.18 | 67.94 | 81.80 | 81.58 | 71.45 |
| NegLabel [12] | 85.64 | 66.30 | 70.38 | 74.74 | 1.56 | 99.59 | 42.59 | **89.89** | 50.04 | 82.63 |
| DualCnst | **82.33** | **69.65** | **67.57** | **76.41** | **1.07** | **99.68** | **42.55** | 88.93 | **48.38** | **83.67** |

## E.2 Other OOD Detection Benchmarks

In Table 14, we present the performance evaluation results using CIFAR-10 and CIFAR-100 [10] as the ID datasets, along with four OOD datasets: iNaturalist [11], SUN [31], Places [32], and Textures [33]. Compared to the NegLabel method, our approach demonstrates significant performance gains. Specifically, on CIFAR-100, DualCnst achieves an average improvement of 23.59% in FPR95 and 9.34% in AUROC. On CIFAR-10, it yields improvements of 7.56% in FPR95 and 1.39% in AUROC. Although DualCnst does not achieve the best performance on CIFAR-10 individually, it outperforms existing methods in terms of overall average performance across both CIFAR-10 and CIFAR-100, highlighting its effectiveness in OOD detection across diverse datasets.

Additionally, in Table 15, we follow the fine-grained dataset setup proposed by EOE [57] and conduct experiments on CUB-200-2011 [43], STANFORD-CARS [44], Food-101 [45], and Oxford-IIIT Pet [46].Under this experimental setting, the four datasets are randomly split into two equal subsets, with one serving as the ID dataset and the other as the OOD dataset. Since NegLabel identifies the most semantically distant candidate labels as potential OOD categories during the OOD label mining process, its performance in fine-grained experiments is relatively suboptimal. In contrast, DualCnst demonstrates superior performance, achieving a 1.48% reduction in FPR95 and an 8.56% improvement in AUROC.

## E.3 Structured Domains: Medical Imaging

We evaluate a highly structured domain by using CheXpert [60] as ID data and sampling 10,000 OOD exemplars from PubMedVision [61]. Medical images present strong, localized morphological cues (e.g., lung fields, cardiomediastinal contours, lesion textures) that can be complementary to textual semantics. Hence, the dual-consistency design is expected to sharpen separability when semantically related categories remain visually distinct.

**Results and interpretation.** As shown in Table 16, DualCnst achieves near-ceiling performance (FPR95= 0.00, AUROC= 99.97), outperforming the text-only NegLabel baseline (FPR95= 0.11, AUROC= 99.92). The improvement, albeit on an already saturated operating point, indicates that incorporating visual similarity to synthesized exemplars reduces residual false positives that persist under text-only scoring. In a domain where subtle visual patterns (e.g., striations, opacities) carry decisive information, the added image-space constraint likely filters spurious semantic matches, consistent with our variance-reduction analysis under weak dependence.

**Implications and caveats.** First, the gains arrive without access to real ID images, preserving data privacy while still leveraging visual priors via synthetic exemplars. Second, because AUROC is

Table 14: Additional empirical results with CIFAR-10 and CIFAR-100 as ID datasets. The **bold** indicates the best performance on each dataset.

| ID Dataset | Method | iNaturalist FPR95↓ | AUROC↑ | SUN FPR95↓ | AUROC↑ | Places FPR95↓ | AUROC↑ | Texture FPR95↓ | AUROC↑ | Average FPR95↓ | AUROC↑ |
|---|---|---|---|---|---|---|---|---|---|---|---|
| | Energy | 60.70 | 82.12 | 53.14 | 86.00 | 58.29 | 82.86 | 62.52 | 77.89 | 58.66 | 82.22 |
| | MaxLogit | 8.99 | 97.85 | **11.81** | **97.36** | 16.74 | 95.55 | 11.54 | 97.60 | **12.27** | 97.09 |
| CIFAR-10 | MCM | 17.87 | 96.75 | 30.78 | 93.17 | 36.57 | 90.78 | 16.38 | 96.44 | 25.40 | 94.29 |
| | NegLabel | 0.55 | **99.84** | 23.31 | 95.50 | 38.70 | 91.53 | 19.33 | 96.65 | 20.47 | 95.88 |
| | DualCnst | **0.42** | 99.83 | 15.23 | 97.07 | 25.46 | 94.17 | **10.55** | **98.00** | 12.91 | **97.27** |
| | Energy | 82.74 | 74.47 | 67.16 | 81.69 | 68.20 | **80.96** | 81.19 | 66.51 | 74.82 | 75.91 |
| | MaxLogit | 67.77 | 81.41 | 63.26 | 80.72 | 65.73 | 80.81 | 62.94 | 82.00 | 64.93 | 81.24 |
| CIFAR-100 | MCM | 97.95 | 67.50 | 97.69 | 60.71 | 98.40 | 61.34 | 90.23 | 73.58 | 96.07 | 65.78 |
| | NegLabel | 13.95 | 96.47 | 86.61 | 69.04 | 91.50 | 62.08 | 70.60 | 80.26 | 65.66 | 76.96 |
| | DualCnst | **2.88** | **99.23** | **49.35** | **84.25** | **60.68** | 79.06 | **55.35** | **82.65** | **42.07** | **86.30** |
| | Energy | 71.72 | 78.30 | 60.15 | 83.84 | 63.25 | 81.91 | 71.86 | 72.20 | 66.74 | 79.06 |
| | MaxLogit | 38.38 | 89.63 | 37.54 | 89.04 | 41.24 | 88.18 | **37.24** | **89.80** | 38.60 | 89.16 |
| Average | MCM | 57.91 | 82.12 | 64.24 | 76.94 | 67.49 | 76.06 | 53.31 | 85.01 | 60.73 | 80.03 |
| | NegLabel | 7.25 | 98.15 | 54.96 | 82.27 | 65.10 | 76.81 | 44.96 | 88.45 | 43.07 | 86.42 |
| | DualCnst | **1.65** | **99.53** | **32.29** | **95.75** | **37.45** | **90.31** | 45.79 | 86.77 | **27.49** | **91.78** |

Table 15: Zero-shot fine-grained OOD detection results. The **black bold** indicates the best performance. The gray indicates that the comparative methods require training or an additional massive auxiliary dataset.

| Method | ID OOD | CUB-100 CUB-100 FPR95↓ | AUROC↑ | Stanford-Cars-98 Stanford-Cars-98 FPR95↓ | AUROC↑ | Food-50 Food-51 FPR95↓ | AUROC↑ | Oxford-Pet-18 Oxford-Pet-19 FPR95↓ | AUROC↑ | Average FPR95↓ | AUROC↑ |
|---|---|---|---|---|---|---|---|---|---|---|---|
| CLIPN | | 73.54 | 74.65 | 53.33 | 82.25 | 43.33 | 88.89 | 53.90 | 86.92 | **56.05** | **83.18** |
| Energy | | 76.13 | 72.11 | 73.78 | 73.82 | 44.95 | 89.97 | 68.51 | 88.34 | 65.84 | 81.06 |
| MaxLogit | | 76.89 | 73.00 | 72.18 | 74.80 | 41.73 | 90.79 | 65.66 | 88.49 | 64.11 | 81.77 |
| MCM | | 83.58 | 67.51 | 83.99 | 68.71 | 43.38 | 91.75 | 63.92 | 84.88 | 68.72 | 78.21 |
| NegLabel | | 82.48 | 68.55 | 79.32 | 70.00 | 37.32 | 92.48 | 66.30 | 88.64 | 66.36 | 79.92 |
| DualCnst | | 77.99 | 72.58 | 78.87 | 70.38 | 36.18 | 92.85 | 66.46 | 88.45 | 64.88 | 81.07 |

Table 16: Performance on medical imaging.

| Method | FPR95↓ / AUROC↑ |
|---|---|
| DualCnst | **0.00 / 99.97** |
| NegLabel | 0.11 / 99.92 |

already near 100, absolute headroom is small; the main benefit is reducing high-TPR false alarms (captured by FPR95). Finally, although medical images are stylistically constrained, domain shifts (scanner, protocol) or prompt/style mismatch in generation could attenuate the benefit; our fusion reduces sensitivity to such shifts by aggregating weakly dependent cues across modalities and layers (Sec. C.2).

# F  Experimental Configuration and Details

## F.1  Details of Mining Potential OOD Labels

Before generating synthetic images, it is crucial to identify effective OOD labels by leveraging ID labels as a reference. Specifically, we define the set of ID labels as $\mathcal{Y}^{\mathrm{id}} = \{y_1, y_2, \ldots, y_K\}$ and collect a pool of nouns and adjectives from open-world resources (e.g., WordNet [62], ConceptNet [63], and Wikipedia Categories[1]) as candidate OOD labels, denoted by $\mathcal{Y}^{\mathrm{c}} = \{\tilde{y}_1, \tilde{y}_2, \ldots, \tilde{y}_C\}$, where $C$ represents the total number of candidates.

To assess the semantic relationship between candidate OOD labels and ID labels, we utilize CLIP's text encoder to extract text embeddings for both sets. The embedding of a candidate OOD label

---

[1] https://dumps.wikimedia.org/

is given by $\tilde{\mathbf{e}}_c = \mathcal{T}(\text{prompt}(\tilde{y}_c))$, while the embedding of an ID label is represented as $\mathbf{e}_k = \mathcal{T}(\text{prompt}(y_k))$. By default, we employ the prompt format `"A photo of a <label>"` to generate these embeddings.

Following the methodology outlined in NegMining [12], we quantify the semantic distance between each candidate OOD label and the ID labels using negative cosine similarity. Specifically, for a given candidate OOD label, we compute its negative cosine similarity with all ID label embeddings, resulting in $K$ similarity scores. The overall semantic distance of an OOD label to the ID label set is then determined as the $\eta$-percentile (default $\eta = 0.05$) of these scores:

$$d_c = \text{percentile}_\eta \left( \{ -\cos(\tilde{\mathbf{e}}_c, \mathbf{e}_k) \}_{k=1}^K \right). \tag{7}$$

After computing distances for all candidate OOD labels, we select the top $M = 10,000$ labels with the greatest distances. The selected OOD label set is defined as:

$$\mathcal{Y}^{\text{ood}} = \text{TopK}\left( \{ d_c \}_{c=1}^C, \mathcal{Y}^{\text{c}}, M \right). \tag{8}$$

During the generation phase, DualCnst utilizes $\mathcal{Y}^{\text{ood}} \cup \mathcal{Y}^{\text{id}}$ as the label space for synthetic image generation. To ensure semantic consistency, it employs stable diffusion to generate images that align with these labels, thereby providing meaningful visual representations to enhance the inference process.

## F.2 Evaluation Metrics

In this study, we adopt the most widely used evaluation metrics in the OOD detection domain, including FPR95 and AUROC [64]. To further assess the effectiveness of the proposed dual consistency method under additional evaluation criteria, we also report AUPR results for CLIP-B/16 in Table 17. The results demonstrate that our dual consistency method achieves superior performance across all evaluation metrics.

Table 17: Performance in terms of AUPR. ID dataset: The experiments are zero-shot OOD detection results with ImageNet-1K as the ID dataset. The **black bold** indicates the best performance. The gray indicates that the comparative methods require training or an additional massive auxiliary dataset.

| Method | OOD Dataset | | | | Average |
| | iNaturalist | SUN | Places | Texture | |
|---|---|---|---|---|---|
| CLIPN | 99.15 | 98.59 | **98.22** | 98.38 | 98.59 |
| Energy | 96.84 | 96.50 | 96.16 | 94.66 | 96.04 |
| MaxLogit | 97.74 | 97.12 | 96.65 | 95.61 | 96.78 |
| MCM | 98.86 | 98.28 | 97.49 | 98.04 | 98.17 |
| NegLabel | 99.80 | 98.79 | 97.76 | 98.08 | 98.61 |
| DualCnst | **99.93** | **99.06** | 98.04 | **98.79** | **98.96** |

## F.3 Experimental Configuration

This paper introduces a dual consistency (DualCnst) method, implemented using Python 3.8 and PyTorch 1.13 library [65], with all experiments conducted on a single NVIDIA RTX A6000 GPU. Prior to experimentation, the proposed method generates synthetic images, with each image requiring approximately 3 seconds for generation. To mitigate redundant computational overhead across multiple runs, we precompute and store the visual features of the generated images.

We systematically benchmark computational efficiency through two key aspects: (1) Temporal cost analysis for synthetic image batch generation across five scales (0, 1k, 10k), with comparative inference latency evaluation against NegLabel [12] (Table 18). DualCnst achieves efficient batch generation in 0.1 seconds (excluding directory initialization overhead), while maintaining competitive inference efficiency (12m 42s vs. 9m 36s for full ImageNet-1k evaluation) with enhanced detection robustness. (2) Performance comparison under varying OOD label quantities. To ensure fair comparison with NegLabel, we adopt its optimal configuration (10,000 OOD labels) for benchmarking. As evidenced in Table 19, our method demonstrates consistent superiority over NegLabel regardless of its OOD label selection size.

For the selection of negative label parameters, we adopt the optimal configuration recommended in NegLabel. All experiments in this study are conducted within the CLIP framework. Unless otherwise specified, we utilize CLIP-B/16 for zero-shot OOD detection. The default hyperparameter settings are as follows: We set $w = 0.1$ and extract intermediate-layer features from the 9th, 10th, and 11th layers of the visual encoder, which are then fused with the final semantic features. The sum-softmax score is employed, with the fusion parameter set to $\alpha = 0.1$ and the temperature parameter to $\tau = 0.01$.

Table 18: Computational cost of DualCnst and NegLabel on ImageNet-1k.

| Method | 0k-ood-image Gen. Time | 1k-ood-image Gen. Time | 10k-ood-image Gen. Time | Inference Time |
|---|---|---|---|---|
| DualCnst | 4m 10s | 9m 42s | 51m 52s | 12m 42s |
| NegLabel | - | - | - | 9m 36s |

Table 19: Performance Comparison Under Varying OOD Label Quantities. The **black bold** indicates the best performance.

| Method(Number of OOD labels) | iNaturalist FPR95↓ | iNaturalist AUROC↑ | SUN FPR95↓ | SUN AUROC↑ | Places FPR95↓ | Places AUROC↑ | Texture FPR95↓ | Texture AUROC↑ | Average FPR95↓ | Average AUROC↑ |
|---|---|---|---|---|---|---|---|---|---|---|
| NegLabel(0k) | 30.91 | 94.61 | 37.59 | 92.57 | 44.69 | 89.77 | 57.77 | 86.11 | 42.74 | 90.77 |
| DualCnst(0k) | 28.56 | 94.77 | 35.23 | 93.38 | 43.99 | 89.79 | 58.72 | 85.82 | 41.63 | 90.94 |
| NegLabel(1k) | 33.76 | 93.56 | 34.34 | 93.07 | 44.51 | 89.49 | 56.51 | 85.73 | 42.28 | 90.46 |
| DualCnst(1k) | 20.75 | 95.96 | 42.09 | 91.40 | 50.61 | 87.48 | 47.70 | 87.43 | 40.29 | 90.57 |
| NegLabel(10k) | 1.29 | 99.65 | 17.60 | 95.89 | 31.91 | **92.13** | 42.15 | 90.51 | 23.24 | 94.55 |
| DualCnst(10k) | **0.98** | **99.70** | **18.13** | 95.66 | **31.77** | 91.81 | **34.91** | **91.73** | **21.45** | **94.72** |

## F.4 Compute Profile and Deployment Details

**Offline pre-generation.** All synthetic exemplars are generated *offline* per label set and reused at inference, so the online overhead is negligible. Using accelerated backbones (e.g., SDXL-Turbo), generating images for 10,000 OOD labels takes ≈55 minutes on a single RTX A6000, i.e., about 0.33 s per label on average. Because this cost is paid once per label set and cached thereafter, latency at serving time is dominated by a single forward pass of the encoders.

**Memory footprint.** We report GPU memory usage under identical configurations and separate *inference* from one-time *generation*.

Table 20: GPU Memory Overhead During Inference and Generation (RTX A6000).

| Method | Inference (MiB) | Generation (MiB) |
|---|---|---|
| DualCnst | 33245 | 9465 |
| NegLabel | 26353 | – |
| MCM | 17827 | – |

**Results and interpretation.** From Table 20, DualCnst uses *33245 MiB* at inference, which is a $+26.2\%$ increase over the text-only NegLabel baseline (26353 MiB), and $+86.6\%$ over MCM (17827 MiB). This overhead is expected: DualCnst keeps both text and image similarity branches active and aggregates multi-layer signals, trading extra activation/feature memory for improved robustness. Crucially, the *generation* footprint (9465 MiB) applies only to the *offline* synthesis stage; it does not affect online serving because exemplars are reused. Thus, for deployment on a single A6000, generation comfortably fits within memory with headroom for modest batching, while inference-time costs remain stable once the exemplars are cached.

**Practical deployment notes.** (i) **Amortization.** Since generation is one-off per label set, the cost amortizes across all future queries and model updates that reuse the same labels. (ii) **Latency.** Online latency is unchanged relative to NegLabel/MCM aside from the added fusion step, which is negligible compared to encoder forward passes. (iii) **Memory controls.** Mixed precision (FP16/BF16), gradient disabling at serve time, and per-layer feature caching reduce inference memory without accuracy loss;

exemplar retrieval can be streamed from pinned host memory when VRAM is tight. (iv) **Scalability.** For larger label vocabularies or multi-tenant settings, shard exemplar banks across GPUs or persist them on disk with an in-memory LRU cache to keep the working set small.

# G   Additional Ablation Studies

## G.1   Efficacy of DualCnst

Our method is not limited to applications with NegLabel. To address this concern, we have conducted ablation studies using the DualCnst method combined with MCM (an OOD detection approach that does not require negative labels) . As shown in Table 21, the results demonstrate that incorporating synthetic image labels can effectively enhance OOD detection performance even without negative labels/images. In addition, we have conducted ablation studies using real OOD samples (Table 22). The results demonstrate that while the proposed method achieves better performance with real samples compared to synthetic ones, the improvement margin is not substantial, demonstrating synthetic samples' effectiveness.

Table 21: Performance comparison of integrating DualCnst with MCM for OOD detection on ImageNet-1k (ID dataset). The **black bold** indicates the best performance.

| Method | OOD Dataset | | | | | | | | Average | |
| | ssb-hard [66] | | NINOC [67] | | iNaturalist | | Texture | | | |
| | FPR95↓ | AUROC↑ | FPR95↓ | AUROC↑ | FPR95↓ | AUROC↑ | FPR95↓ | AUROC↑ | FPR95↓ | AUROC↑ |
|---|---|---|---|---|---|---|---|---|---|---|
| MCM | 89.43 | **64.15** | 82.70 | 69.85 | 30.91 | 94.61 | **57.77** | **86.11** | 65.20 | 78.68 |
| MCM+DualCnst | **88.69** | 63.66 | **78.96** | **74.05** | **28.56** | **94.77** | 58.72 | 85.82 | **63.73** | **79.58** |

Table 22: Experimental comparison between real OOD samples and synthetic OOD samples, with ImageNet-1k as the ID dataset. The **black bold** indicates the best performance.

| Source of OOD images | OOD Dataset | | | | | | | | Average | |
| | iNaturalist | | SUN | | Places | | Texture | | | |
| | FPR95↓ | AUROC↑ | FPR95↓ | AUROC↑ | FPR95↓ | AUROC↑ | FPR95↓ | AUROC↑ | FPR95↓ | AUROC↑ |
|---|---|---|---|---|---|---|---|---|---|---|
| Synthetic OOD samples | **1.29** | **99.65** | 17.60 | 95.89 | 31.91 | 92.13 | 42.15 | 90.51 | 23.24 | 94.55 |
| Real OOD samples | 1.73 | 99.28 | **6.55** | **98.67** | **18.51** | **95.69** | **22.64** | **95.35** | **12.36** | **97.25** |

## G.2   Effect of Prompt Engineering

We evaluate NegLabel with different prompt template on the ImageNet-1k benchmark, as shown in Table 23. The first four rows are selected from the 80 prompt templates proposed by CLIP [8]. The last two rows are NegLabel designed prompt templates.According to the results in the table, DualCnst significantly outperforms NegLabel in OOD detection performance across multiple prompts.

## G.3   Vision Backbone

This section explores the performance of DualCnst using different CLIP vision encoders.

Table 24 presents the results for ImageNet-1K (ID) with various CLIP vision encoders, including ViT-B/32[2], ViT-L/14[3], RN50[4], RN50x4, RN50x16, and RN101. Across all tested encoders, DualCnst achieves the highest performance. Specifically, compared to ViT-B/16, using ViT-L/14 results in an improvement of 2.33% in FPR95 and 0.37% in AUROC. Furthermore, DualCnst outperforms both zero-shot and fine-tuning methods in OOD detection, achieving the best results in terms of FPR95 and AUROC when utilizing ViT-L/14.

---

[2]https://huggingface.co/openai/clip-vit-base-patch32
[3]https://huggingface.co/openai/clip-vit-large-patch14
[4]https://github.com/openai/CLIP

Table 23: Performance Comparison of Different Prompts. The **bold** indicates the best performance on each dataset.

| Prompt | Method | iNaturalist | | SUN | | Places | | Texture | | Average | |
|---|---|---|---|---|---|---|---|---|---|---|---|
| | | FPR95↓ | AUROC↑ | FPR95↓ | AUROC↑ | FPR95↓ | AUROC↑ | FPR95↓ | AUROC↑ | FPR95↓ | AUROC↑ |
| a photo of a <label> | NegLabel | 31.89 | 94.33 | 50.63 | 91.54 | 64.21 | 87.37 | 84.95 | 73.17 | 57.92 | 86.60 |
| | DualCnst | 19.67 | 96.02 | 49.54 | 91.52 | 58.74 | 87.66 | 71.29 | 79.70 | 49.81 | 88.72 |
| a photo of many <label> | NegLabel | 71.03 | 86.84 | 79.96 | 85.94 | 77.04 | 83.94 | 86.10 | 73.01 | 78.53 | 82.43 |
| | DualCnst | 56.51 | 90.70 | 79.23 | 86.67 | 76.48 | 83.77 | 71.72 | 80.52 | 70.98 | 85.41 |
| a photo of the <label> | NegLabel | 45.53 | 92.06 | 43.81 | 91.83 | 55.91 | 88.13 | 74.73 | 78.32 | 55.00 | 87.59 |
| | DualCnst | 39.37 | 92.94 | 43.30 | 92.02 | 53.08 | 87.99 | 63.32 | 83.64 | 49.77 | 89.15 |
| a low resolution photo of a <label> | NegLabel | 51.39 | 90.35 | 69.74 | 88.23 | 75.01 | 84.37 | 88.58 | 72.28 | 71.18 | 83.81 |
| | DualCnst | 38.07 | 93.23 | 66.37 | 88.76 | 70.13 | 85.36 | 71.83 | 80.07 | 61.60 | 86.86 |
| <label> | NegLabel | 52.77 | 91.15 | 41.93 | 92.40 | 49.13 | 89.45 | 68.65 | 82.61 | 53.12 | 88.90 |
| | DualCnst | 28.35 | 94.66 | 42.36 | 91.58 | 50.67 | 88.37 | 52.34 | 87.78 | 43.43 | 90.60 |
| the nice <label> | NegLabel | 1.91 | 99.49 | 20.53 | 95.49 | 35.59 | 91.64 | 43.56 | 90.22 | 25.40 | 94.21 |
| | DualCnst | **0.98** | **99.70** | **18.13** | **95.66** | **31.77** | **91.81** | **34.91** | **91.73** | **21.45** | **94.72** |

Table 24: Prompt ensembling for text input using different backbones. The ID dataset is ImageNet-1K. The **black bold** indicates the best performance.

| Method | iNaturalist | | SUN | | Places | | Texture | | Average | |
|---|---|---|---|---|---|---|---|---|---|---|
| | FPR95↓ | AUROC↑ | FPR95↓ | AUROC↑ | FPR95↓ | AUROC↑ | FPR95↓ | AUROC↑ | FPR95↓ | AUROC↑ |
| Energy (ViT-B/16) | 79.75 | 83.75 | 79.81 | 83.21 | 70.28 | 83.95 | 88.23 | 71.51 | 79.52 | 80.60 |
| MaxLogit (ViT-B/16) | 67.24 | 87.31 | 66.14 | 86.36 | 61.09 | 85.96 | 80.83 | 76.01 | 68.83 | 83.91 |
| MCM (ViT-B/16) | 40.33 | 92.75 | 35.43 | 92.78 | 44.08 | 89.60 | 54.41 | 87.10 | 43.56 | 90.56 |
| NegLabel (ViT-B/16) | 1.91 | 99.49 | 20.53 | 95.49 | 35.59 | 91.64 | 43.56 | 90.22 | 25.40 | 94.21 |
| DualCnst (ViT-B/16) | **1.29** | **99.65** | **17.60** | **95.89** | **31.91** | **92.13** | **42.15** | **90.51** | **23.24** | **94.55** |
| Energy (ViT-B/32) | 89.22 | 79.15 | 81.01 | 81.62 | 61.22 | 87.20 | 87.64 | 71.36 | 79.77 | 79.83 |
| MaxLogit (ViT-B/32) | 79.45 | 83.75 | 68.89 | 84.85 | 52.30 | 88.60 | 79.88 | 75.29 | 70.13 | 83.12 |
| MCM (ViT-B/32) | 49.81 | 91.37 | 40.31 | 91.80 | 42.94 | 90.08 | 59.33 | 85.32 | 48.10 | 89.64 |
| NegLabel (ViT-B/32) | 3.73 | 99.11 | 22.48 | 95.27 | 34.94 | 91.72 | **50.51** | **88.57** | 27.92 | 93.67 |
| DualCnst (ViT-B/32) | **3.10** | **99.27** | **18.93** | **95.87** | **32.43** | **92.13** | 53.56 | 88.10 | **27.01** | **93.84** |
| Energy (ViT-L/14) | 79.20 | 85.29 | 76.83 | 84.68 | 65.62 | 87.59 | 87.23 | 70.14 | 77.22 | 81.93 |
| MaxLogit (ViT-L/14) | 63.06 | 89.02 | 60.26 | 88.29 | 52.51 | 89.65 | 80.66 | 73.96 | 64.12 | 85.23 |
| MCM (ViT-L/14) | 31.63 | 94.43 | 23.64 | 94.99 | 30.99 | 92.79 | 57.77 | 85.19 | 36.01 | 91.85 |
| NegLabel (ViT-L/14) | 1.77 | 99.53 | 22.33 | 95.63 | 32.22 | 93.01 | 42.92 | 89.71 | 24.81 | 94.47 |
| DualCnst (ViT-L/14) | **1.33** | **99.70** | **19.54** | **96.06** | **26.55** | **93.72** | **42.48** | **89.87** | **22.48** | **94.84** |
| Energy (RN50) | 94.75 | 75.56 | 86.24 | 81.39 | 86.42 | 78.68 | 92.98 | 69.87 | 90.10 | 76.38 |
| MaxLogit (RN50) | 86.45 | 81.21 | 74.56 | 84.31 | 78.15 | 81.10 | 86.45 | 74.61 | 81.40 | 80.31 |
| MCM (RN50) | 45.42 | 91.50 | 43.33 | 91.40 | 55.92 | 86.73 | 55.92 | 86.68 | 50.15 | 89.08 |
| NegLabel (RN50) | 2.88 | 99.24 | 26.51 | 94.54 | 42.60 | 89.72 | **50.80** | 88.40 | 30.70 | 92.97 |
| DualCnst (RN50) | **1.81** | **99.51** | **20.75** | **95.39** | **35.10** | **91.13** | 51.19 | **88.90** | **27.21** | **93.73** |
| Energy (RN50x4) | 85.55 | 81.25 | 80.13 | 84.81 | 68.84 | 85.40 | 92.09 | 69.28 | 81.65 | 80.19 |
| MaxLogit (RN50x4) | 74.51 | 85.14 | 65.51 | 87.61 | 58.86 | 87.26 | 84.47 | 74.81 | 70.84 | 83.70 |
| MCM (RN50x4) | 48.00 | 90.86 | 33.81 | 93.14 | 42.90 | 89.93 | 52.16 | 87.44 | 44.22 | 90.34 |
| NegLabel (RN50x4) | 2.14 | 99.49 | 17.61 | 96.25 | 30.67 | 92.59 | 50.71 | 88.72 | 25.28 | 94.26 |
| DualCnst (RN50x4) | **1.58** | **99.62** | **16.89** | **96.27** | **29.04** | **92.63** | **47.29** | **89.60** | **23.70** | **94.53** |
| Energy (RN50x16) | 73.44 | 86.95 | 65.15 | 88.97 | 73.74 | 83.97 | 84.43 | 76.11 | 74.19 | 84.00 |
| MaxLogit (RN50x16) | 62.10 | 89.05 | 52.35 | 90.45 | 64.74 | 85.69 | 75.66 | 79.37 | 63.71 | 86.14 |
| MCM (RN50x16) | 43.02 | 91.69 | 34.24 | 93.27 | 46.96 | 89.27 | 51.93 | 87.94 | 44.04 | 90.54 |
| NegLabel (RN50x16) | 2.00 | 99.48 | 29.11 | 94.18 | 48.14 | 88.85 | **38.74** | **91.23** | 29.50 | 93.43 |
| DualCnst (RN50x16) | **1.22** | **99.66** | **19.42** | **95.80** | **34.51** | **91.73** | 39.34 | 91.17 | **23.62** | **94.59** |
| Energy (RN101) | 97.82 | 71.11 | 87.81 | 81.10 | 85.43 | 77.92 | 95.96 | 62.32 | 91.75 | 73.11 |
| MaxLogit (RN101) | 92.65 | 77.38 | 74.77 | 84.67 | 75.96 | 81.30 | 90.90 | 68.66 | 83.57 | 78.00 |
| MCM (RN101) | 60.90 | 88.14 | 39.37 | 91.96 | 48.62 | 88.08 | 59.49 | 85.34 | 52.09 | 88.38 |
| NegLabel (RN101) | 2.35 | 99.42 | 21.84 | 95.45 | 41.98 | 90.08 | **53.95** | **87.68** | 30.03 | 93.16 |
| DualCnst (RN101) | 2.56 | 99.36 | **18.93** | **95.88** | **37.52** | **90.89** | 56.03 | 86.88 | **28.76** | **93.26** |

## G.4 Generative Models

To evaluate the effectiveness of DualCnst, different generative models are employed. Table 25 compares the performance of Stable Diffusion v1.5[5] and v2.1[6] and Hunyuan-DiT[7] and sdxl-turbo[8] using ImageNet-1k as the ID dataset. The performance improvements vary depending on the quality of generated synthetic images. Notably, sdxl-turbo achieves the highest synthetic image quality, leading to the most significant performance gain, whereas Stable Diffusion v1.5 yields the lowest-quality generations with minimal improvement compared to other models. Nevertheless, both still surpass our baseline. In terms of computational efficiency, Hunyuan-DiT exhibits the longest generation time, while sdxl-turbo demonstrates the fastest inference speed. Stable Diffusion v1.5 and v2.1 show comparable computational latency.

Table 25: The impact of randomness under different random seeds is examined, with ImageNet-1k as the ID dataset. The **black bold** indicates the best performance.

| Generative Models | OOD Dataset | | | | | | | | Average | |
| | iNaturalist | | SUN | | Places | | Texture | | | |
| | FPR95↓ | AUROC↑ | FPR95↓ | AUROC↑ | FPR95↓ | AUROC↑ | FPR95↓ | AUROC↑ | FPR95↓ | AUROC↑ |
|---|---|---|---|---|---|---|---|---|---|---|
| stable diffusion v1.5 | 1.27 | 99.65 | 17.30 | 95.94 | 31.61 | 92.15 | 42.91 | 90.32 | 23.27 | 94.51 |
| stable diffusion v2.1 | 1.42 | 99.63 | 17.93 | 95.86 | 32.22 | 92.13 | **39.11** | 90.97 | 22.67 | 94.65 |
| Hunyuan-DiT | 1.51 | 99.62 | 17.55 | **95.95** | 31.24 | **92.18** | 42.46 | 90.37 | 23.19 | 94.53 |
| stable diffusion XL-Turbo | **1.25** | **99.66** | **16.91** | 95.94 | **30.73** | 92.17 | 39.20 | **90.98** | **22.02** | **94.69** |

## G.5 Encoder Layer

An ablation study was conducted to evaluate the effectiveness of DualCnst using different layers of CLIP's ViT-B/16 encoder. As demonstrated in Table 26, we present the results of SDXL-Turbo under various layer combinations: (1st, 2nd, 3rd), (4th, 5th, 6th), (7th, 8th, 9th), (9th, 10th, 11th), (3rd, 6th, 9th), and all layers. For each combination, multiple weighting coefficients $w$ (0.05, 0.1, 0.15, 0.25) were systematically investigated to identify the optimal balance between pixel-level fidelity and semantic consistency.

The results indicate that an equal weight distribution is not necessarily optimal across different layers. For instance, when using the (1st, 2nd, 3rd) layers, setting $w = 0.05$ yields the best performance, as the lower layers primarily capture edge-related features, requiring stronger semantic guidance. In contrast, for the (9th, 10th, 11th) layers, which encode more localized details—such as the fur and ears of a wolf or the stripes of a zebra—assigning a higher weight to visual features leads to improved performance within the DualCnst framework.

## G.6 Fusion Parameter $\alpha$ of Dual Consistency

This section presents a comprehensive ablation study on the fusion parameter $\alpha$ in the dual consistency method. Experiments are conducted using ImageNet-1k, CIFAR-10, and CIFAR-100 as ID datasets, with iNaturalist, SUN, Places, and Textures serving as OOD datasets. Additionally, experiments are performed by alternately designating ImageNet-10 and ImageNet-20 as ID and OOD datasets.

All experiments utilize the ViT-B/16 visual encoder with selected layers (7th, 8th, 9th) and a fixed weight parameter of $w = 0.15$. As shown in Table 29, the optimal $\alpha$ value varies across different OOD datasets for ImageNet-1k. Specifically, the best results are obtained with $\alpha = 0.1$ for SUN and Places, and $\alpha = 0.2$ for iNaturalist and Textures. In the main results, we select $\alpha = 0.2$ for our presented results. Notably, when $\alpha = 0$, DualCnst reduces to NegLabel.

Table 27 and Table 28 present the results for the CIFAR datasets, where DualCnst consistently outperforms NegLabel. Furthermore, as shown in Table 30, when the ID and OOD datasets exhibit semantic similarities, integrating DualCnst leads to notable performance improvements.

---

[5] https://github.com/CompVis/stable-diffusion
[6] https://huggingface.co/stabilityai/stable-diffusion-2-1
[7] https://huggingface.co/Tencent-Hunyuan/HunyuanDiT-v1.2-Diffusers
[8] https://huggingface.co/stabilityai/sdxl-turbo

Table 26: Using **SDXL-Turbo** different encoder layers and weights. The ID class labels are the same as ImageNet-1k. The **black bold** indicates the best performance.

| Layer | $w$ | iNaturalist | | SUN | | Places | | Texture | | Average | |
|---|---|---|---|---|---|---|---|---|---|---|---|
| | | **FPR95↓** | **AUROC↑** | **FPR95↓** | **AUROC↑** | **FPR95↓** | **AUROC↑** | **FPR95↓** | **AUROC↑** | **FPR95↓** | **AUROC↑** |
| (1st, 2nd, 3rd) | 0.05 | **1.29** | **99.67** | 17.49 | **95.86** | **31.36** | **92.05** | 40.82 | 90.83 | **22.74** | **94.60** |
| | 0.10 | **1.29** | **99.67** | **17.46** | 95.82 | 31.51 | 92.01 | 41.52 | 90.74 | 22.95 | 94.56 |
| | 0.15 | 1.28 | 99.67 | 17.59 | 95.76 | 31.86 | 91.94 | 41.76 | 90.63 | 23.12 | 94.50 |
| | 0.25 | 1.30 | 99.65 | 18.44 | 95.56 | 33.26 | 91.67 | 43.42 | 90.25 | 24.11 | 94.28 |
| (4th, 5th, 6th) | 0.05 | 1.24 | 99.67 | 17.49 | **95.90** | 31.23 | 92.11 | 39.75 | 90.96 | 22.43 | 94.66 |
| | 0.10 | 1.21 | **99.68** | 17.42 | 95.89 | **31.09** | **92.13** | 39.38 | **90.98** | 22.27 | **94.67** |
| | 0.15 | **1.13** | **99.68** | **17.40** | 95.85 | 31.16 | 92.10 | **39.11** | 90.93 | **22.20** | 94.64 |
| | 0.25 | 1.17 | 99.65 | 18.33 | 95.67 | 32.68 | 91.89 | 39.73 | 90.65 | 22.98 | 94.46 |
| (7th, 8th, 9th) | 0.05 | 1.25 | 99.67 | **17.23** | 95.91 | **30.86** | 92.11 | 39.29 | 91.02 | 22.16 | 94.68 |
| | 0.10 | 1.23 | 99.67 | 17.33 | **95.92** | 30.91 | **92.14** | 38.94 | 91.14 | 22.10 | **94.72** |
| | 0.15 | **0.98** | **99.70** | 18.13 | 95.66 | 31.77 | 91.81 | **34.91** | **91.73** | **21.45** | **94.72** |
| (9th, 10th, 11th) | 0.05 | 1.27 | 99.66 | 17.03 | 95.92 | 30.76 | 92.12 | 39.61 | 90.94 | 22.17 | 94.66 |
| | 0.10 | 1.25 | 99.66 | 16.91 | 95.94 | **30.73** | 92.17 | 39.20 | 90.98 | 22.02 | 94.69 |
| | 0.15 | **1.23** | **99.66** | **16.79** | **95.96** | 30.74 | **92.20** | **39.13** | **91.00** | **21.97** | **94.70** |
| | 0.25 | 1.24 | 99.65 | 16.88 | 95.93 | 31.36 | 92.19 | 39.33 | 90.95 | 22.20 | 94.68 |
| (3rd, 6th, 9th) | 0.05 | 1.24 | 99.67 | 17.35 | 95.90 | 31.10 | 92.11 | 39.80 | 90.94 | 22.37 | 94.65 |
| | 0.10 | 1.23 | 99.67 | 17.26 | 95.92 | **31.06** | 92.14 | 39.77 | 90.97 | 22.33 | 94.68 |
| | 0.15 | 1.22 | 99.67 | **17.01** | **95.91** | 31.12 | **92.15** | **39.36** | **90.97** | **22.18** | 94.68 |
| | 0.25 | **1.15** | **99.67** | 17.27 | 95.85 | 31.76 | 92.10 | 39.80 | 90.87 | 22.50 | 94.62 |
| all layer | 0.01 | 1.25 | 99.67 | 17.36 | 95.89 | **31.07** | 92.09 | 39.80 | 90.93 | 22.37 | 94.64 |
| | 0.02 | 1.24 | 99.67 | 17.43 | **95.90** | 31.16 | **92.11** | 39.91 | **90.96** | 22.44 | 94.66 |
| | 0.05 | 1.19 | **99.68** | **17.05** | 95.88 | 31.13 | 92.11 | **39.54** | 90.95 | **22.23** | **94.66** |
| | 0.08 | **1.13** | 99.66 | 17.62 | 95.76 | 32.32 | 91.97 | 39.91 | 90.76 | 22.75 | 94.54 |

Table 27: An ablation study on the fusion parameter $\alpha$ for cifar10. The **black bold** indicates the best performance.

| $\alpha$ | iNaturalist | | SUN | | Places | | Texture | | Average | |
|---|---|---|---|---|---|---|---|---|---|---|
| | **FPR95↓** | **AUROC↑** | **FPR95↓** | **AUROC↑** | **FPR95↓** | **AUROC↑** | **FPR95↓** | **AUROC↑** | **FPR95↓** | **AUROC↑** |
| 0 | 0.55 | 99.84 | 23.31 | 95.5 | 38.7 | 91.53 | 19.33 | 96.65 | 20.47 | 95.88 |
| 0.1 | 0.35 | 99.85 | 17.57 | 96.46 | 30.33 | 93.10 | 12.75 | 97.59 | 15.25 | 96.75 |
| 0.2 | **0.33** | **99.85** | 15.38 | 96.88 | 26.43 | 93.82 | 10.80 | 97.93 | 13.23 | 97.12 |
| 0.3 | 0.42 | 99.83 | **15.23** | 97.07 | 25.46 | 94.17 | 10.55 | **98.00** | **12.91** | 97.27 |
| 0.4 | 0.52 | 99.80 | 15.30 | **97.12** | **25.33** | **94.30** | **10.53** | 97.94 | 12.92 | **97.29** |
| 0.5 | 0.67 | 99.75 | 15.75 | 97.10 | 25.55 | 94.31 | 11.01 | 97.84 | 13.25 | 97.25 |
| 0.6 | 0.89 | 99.67 | 16.18 | 97.04 | 26.17 | 94.24 | 11.49 | 97.71 | 13.68 | 97.17 |
| 0.7 | 1.39 | 99.55 | 16.74 | 96.95 | 26.89 | 94.14 | 11.95 | 97.59 | 14.24 | 97.06 |
| 0.8 | 1.97 | 99.38 | 17.14 | 96.86 | 27.49 | 94.03 | 12.27 | 97.47 | 14.72 | 96.94 |
| 0.9 | 3.13 | 99.16 | 17.71 | 96.76 | 27.87 | 93.91 | 12.68 | 97.37 | 15.35 | 96.80 |
| 1 | 4.74 | 98.88 | 18.27 | 96.66 | 28.26 | 93.79 | 13.14 | 97.26 | 16.10 | 96.65 |

Table 28: An ablation study on the fusion parameter $\alpha$ for cifar100. The **black bold** indicates the best performance.

| $\alpha$ | iNaturalist | | SUN | | Places | | Texture | | Average | |
|---|---|---|---|---|---|---|---|---|---|---|
| | **FPR95↓** | **AUROC↑** | **FPR95↓** | **AUROC↑** | **FPR95↓** | **AUROC↑** | **FPR95↓** | **AUROC↑** | **FPR95↓** | **AUROC↑** |
| 0 | 13.95 | 96.47 | 86.61 | 69.04 | 91.5 | 62.08 | 70.6 | 80.26 | 65.66 | 76.96 |
| 0.1 | 11.63 | 97.18 | 83.36 | 70.68 | 89.27 | 64.39 | 61.79 | 83.27 | 61.51 | 78.88 |
| 0.2 | 9.55 | 97.72 | 80.42 | 72.56 | 87.11 | 66.52 | 57.20 | 84.67 | 58.57 | 80.37 |
| 0.3 | 7.51 | 98.11 | 75.99 | 74.40 | 83.68 | 68.46 | 54.77 | **85.01** | 55.49 | 81.50 |
| 0.4 | 6.40 | 98.42 | 72.84 | 76.14 | 79.90 | 70.26 | 55.48 | 84.72 | 53.65 | 82.39 |
| 0.5 | 5.29 | 98.66 | 67.84 | 77.83 | 75.92 | 72.01 | 55.62 | 84.22 | 51.17 | 83.18 |
| 0.6 | 4.40 | 98.85 | 63.12 | 79.45 | 72.40 | 73.70 | 55.12 | 83.74 | 48.76 | 83.94 |
| 0.7 | 3.80 | 99.00 | 58.51 | 80.94 | 68.65 | 75.30 | 54.96 | 83.38 | 46.48 | 84.66 |
| 0.8 | 3.33 | 99.11 | 54.69 | 82.25 | 65.38 | 76.75 | **54.73** | 83.10 | 44.53 | 85.30 |
| 0.9 | 2.97 | 99.18 | 51.49 | 83.35 | 62.70 | 78.00 | 55.12 | 82.86 | 43.07 | 85.85 |
| 1 | **2.88** | **99.23** | **49.35** | **84.25** | **60.68** | **79.06** | 55.35 | 82.65 | **42.07** | **86.30** |

Table 29: An ablation study on the fusion parameter $\alpha$ for ImageNet-1k. The **black bold** indicates the best performance.

| $\alpha$ | iNaturalist FPR95↓ | iNaturalist AUROC↑ | SUN FPR95↓ | SUN AUROC↑ | Places FPR95↓ | Places AUROC↑ | Texture FPR95↓ | Texture AUROC↑ | Average FPR95↓ | Average AUROC↑ |
|---|---|---|---|---|---|---|---|---|---|---|
| 0 | 1.91 | 99.49 | 20.53 | 95.49 | 35.59 | 91.64 | 43.56 | 90.22 | 25.40 | 94.21 |
| 0.1 | 1.23 | 99.66 | **16.79** | **95.96** | **30.74** | **92.20** | 39.13 | 91.00 | 21.97 | 94.70 |
| 0.2 | **0.98** | **99.70** | 18.13 | 95.66 | 31.77 | 91.81 | 34.91 | 91.73 | **21.45** | **94.72** |
| 0.3 | 1.00 | 99.68 | 21.01 | 95.13 | 33.80 | 91.17 | **34.06** | **91.80** | 22.47 | 94.45 |
| 0.4 | 1.27 | 99.63 | 24.70 | 94.42 | 36.79 | 90.39 | 34.31 | 91.64 | 24.27 | 94.02 |
| 0.5 | 1.65 | 99.55 | 28.39 | 93.61 | 39.76 | 89.54 | 35.76 | 91.34 | 26.39 | 93.51 |
| 0.6 | 2.12 | 99.45 | 31.79 | 92.75 | 42.59 | 88.68 | 37.82 | 90.95 | 28.58 | 92.96 |
| 0.7 | 2.80 | 99.32 | 35.10 | 91.87 | 44.95 | 87.82 | 39.27 | 90.50 | 30.53 | 92.38 |
| 0.8 | 3.63 | 99.18 | 38.41 | 90.98 | 46.84 | 86.98 | 41.03 | 89.99 | 32.48 | 91.78 |
| 0.9 | 4.33 | 99.02 | 40.90 | 90.13 | 48.76 | 86.16 | 42.23 | 89.44 | 34.06 | 91.19 |
| 1 | 5.12 | 98.85 | 43.32 | 89.32 | 50.37 | 85.38 | 43.63 | 88.86 | 35.61 | 90.60 |

Table 30: An ablation study on the parameter $\alpha$, alternating ImageNet10 and ImageNet20 as ID and OOD datasets. The **black bold** indicates the best performance.

| $\alpha$ | ID: ImageNet-10 / OOD: ImageNet-20 FPR95↓ | AUROC↑ | ID: ImageNet-20 / OOD: ImageNet-10 FPR95↓ | AUROC↑ | Average FPR95↓ | Average AUROC↑ |
|---|---|---|---|---|---|---|
| 0.0 | 5.10 | 98.86 | 17.60 | 97.04 | 11.35 | 97.95 |
| 0.1 | **2.20** | 98.77 | 13.60 | 97.61 | 8.15 | **98.31** |
| 0.2 | 3.10 | **98.98** | 12.20 | **97.44** | 7.65 | 98.21 |
| 0.3 | 4.00 | 98.89 | 18.60 | 97.06 | 11.30 | 97.98 |
| 0.4 | 4.90 | 98.72 | 18.40 | 97.09 | 11.65 | 97.91 |
| 0.5 | 5.30 | 98.54 | 18.00 | 97.12 | 11.65 | 97.83 |
| 0.6 | 6.40 | 98.33 | 17.80 | 97.16 | 12.10 | 97.75 |
| 0.7 | 8.60 | 98.07 | 16.80 | 97.21 | 12.70 | 97.64 |
| 0.8 | 10.30 | 97.78 | 15.40 | 97.41 | 12.85 | 97.60 |
| 0.9 | 12.60 | 97.47 | 13.60 | 97.65 | 13.10 | 97.56 |
| 1 | 13.10 | 97.14 | 48.00 | 97.65 | 30.55 | 97.40 |

## G.7 Impact of Noise and Distribution Shifts in Synthetic Images on OOD Detection Performance

While synthetic images may contain noise, style variations, or artifacts that could potentially mislead OOD detection models, our score function incorporates multi-level features from pixel to semantic levels. Thus, even when pixel-level features are affected by such deviations, language-level features remain effective. To validate this, we generated oil painting-style images to simulate style shifts. As shown in Table 31, our method maintains robust performance despite this variation.

Table 31: Experimental comparison under generated image style shifts. ID dataset: ImageNet-1k. The **black bold** indicates the best performance.

| Image Style (Method) | iNaturalist FPR95↓ | iNaturalist AUROC↑ | SUN FPR95↓ | SUN AUROC↑ | Places FPR95↓ | Places AUROC↑ | Texture FPR95↓ | Texture AUROC↑ | Average FPR95↓ | Average AUROC↑ |
|---|---|---|---|---|---|---|---|---|---|---|
| Natural Images (DualCnst) | **1.29** | **99.65** | **17.60** | **95.89** | **31.91** | **92.13** | **42.15** | **90.50** | **23.24** | **94.55** |
| Oil Painting Images (DualCnst) | 1.31 | 99.65 | 17.73 | 95.79 | 32.18 | 92.00 | 44.10 | 90.04 | 23.83 | 94.37 |
| NegLabel | 1.91 | 99.49 | 20.53 | 95.49 | 35.59 | 91.64 | 43.56 | 90.22 | 25.40 | 94.21 |

## G.8 The Randomness of Stable Diffusion

An ablation study is conducted to assess the impact of Stable Diffusion's randomness on the effectiveness of DualCnst in generating synthetic images. Specifically, synthetic images are generated using three distinct random seeds with SD1.5 and SDXL-Turbo, followed by performance evaluation on ImageNet-1k. The experimental configuration employs the ViT-B/16 visual encoder with the (9th, 10th, 11th) layer combination, alongside fixed hyperparameters $w = 0.1$ and $\alpha = 0.1$.

As demonstrated in Tables 32 (SDXL-Turbo) and 33 (SD1.5), the detection performance exhibits remarkable consistency across varying random seeds, demonstrating that the generative randomness inherent in Stable Diffusion exerts negligible influence on the operational efficacy of DualCnst.

Table 32: Evaluate SDXL-Turbo's performance variations under different random seeds using ImageNet-1k as the ID dataset. The **black bold** indicates the best performance.

| Random | OOD Dataset | | | | | | | | Average | |
| | iNaturalist | | SUN | | Places | | Texture | | | |
| | FPR95↓ | AUROC↑ | FPR95↓ | AUROC↑ | FPR95↓ | AUROC↑ | FPR95↓ | AUROC↑ | FPR95↓ | AUROC↑ |
| --- | --- | --- | --- | --- | --- | --- | --- | --- | --- | --- |
| random 1 | **0.98** | **99.70** | 18.13 | 95.66 | 31.77 | 91.81 | **34.91** | **91.73** | 21.45 | 94.72 |
| random 2 | 1.25 | 99.66 | **16.62** | **95.96** | 30.84 | 92.19 | 39.29 | 91.02 | 22.00 | 94.71 |
| random 3 | 1.26 | 99.66 | 16.74 | 96.00 | **30.82** | **92.23** | 39.70 | 90.89 | 22.13 | 94.69 |

Table 33: Evaluate Stable Diffusion v1.5's performance variations under different random seeds using ImageNet-1k as the ID dataset. The **black bold** indicates the best performance.

| Random | OOD Dataset | | | | | | | | Average | |
| | iNaturalist | | SUN | | Places | | Texture | | | |
| | FPR95↓ | AUROC↑ | FPR95↓ | AUROC↑ | FPR95↓ | AUROC↑ | FPR95↓ | AUROC↑ | FPR95↓ | AUROC↑ |
| --- | --- | --- | --- | --- | --- | --- | --- | --- | --- | --- |
| random 1 | 1.27 | 99.65 | **17.30** | **95.94** | **31.61** | **92.15** | 42.91 | 90.32 | 23.27 | 94.51 |
| random 2 | 1.29 | 99.65 | 17.60 | 95.89 | 31.91 | 92.13 | 42.15 | **90.51** | **23.24** | **94.55** |
| random 3 | **1.26** | **99.66** | 18.16 | 95.64 | 32.26 | 91.90 | **41.90** | 90.33 | 23.39 | 94.38 |

Table 34: Additional ablation studies on score functions. The **bold** indicates the best performance on each dataset.

| Score Funtion | OOD Dataset | | | | | | | | Average | |
| | iNaturalist | | SUN | | Places | | Texture | | | |
| | FPR95↓ | AUROC↑ | FPR95↓ | AUROC↑ | FPR95↓ | AUROC↑ | FPR95↓ | AUROC↑ | FPR95↓ | AUROC↑ |
| --- | --- | --- | --- | --- | --- | --- | --- | --- | --- | --- |
| $S_{MAX}$ | 100.00 | 83.00 | 100.00 | 82.16 | 100.00 | 80.62 | 100.00 | 80.28 | 100.00 | 81.51 |
| $S_{Energy}$ | 1.98 | 99.42 | 20.26 | 95.52 | 35.54 | 91.49 | 45.69 | 89.96 | 25.87 | 94.10 |
| $S_{MaxLogit}$ | 6.26 | 98.58 | 29.54 | 93.79 | 43.18 | 89.55 | 50.78 | 87.95 | 32.44 | 92.47 |
| $S_{DualCnst}$ | **0.98** | **99.70** | **18.13** | **95.66** | **31.77** | **91.81** | **34.91** | **91.73** | **21.45** | **94.72** |

## G.9   Open-Vocabulary Scaling via OOD Label Clustering

We simulate open-vocabulary conditions by clustering 10,145 fine-grained OOD categories into 136 coarse clusters and compare performance with the unclustered (10,000-label) setting. Heavy clustering reduces label granularity, which is expected to weaken the representational diversity of negative exemplars.

As shown in Table 35, clustering (coarsening) the OOD label vocabulary leads to a pronounced degradation across all benchmarks: the average FPR95 nearly doubles (from 21.45 to 41.78), while AUROC drops by more than 4 points on average. The decline is especially severe on **SUN** and **Places**, where visual and semantic diversity among categories is high and coarse labels (e.g., "indoor scenes") fail to capture subtle inter-class distinctions.

Table 35: Effect of clustering OOD labels (FPR95↓ / AUROC↑). The **black bold** highlights the best performance in each column.

| Method | OOD Dataset | | | | | | | | Average | |
| | iNaturalist | | SUN | | Places | | Texture | | | |
| | FPR95↓ | AUROC↑ | FPR95↓ | AUROC↑ | FPR95↓ | AUROC↑ | FPR95↓ | AUROC↑ | FPR95↓ | AUROC↑ |
| --- | --- | --- | --- | --- | --- | --- | --- | --- | --- | --- |
| DualCnst (10,000 labels) | **0.98** | **99.70** | **18.13** | **95.66** | **31.77** | **91.81** | **34.91** | **91.73** | **21.45** | **94.72** |
| DualCnst (clustered: 136 clusters) | 12.13 | 97.58 | 54.02 | 88.48 | 57.06 | 87.02 | 43.92 | 89.40 | 41.78 | 90.62 |

These results empirically validate that **fine-grained negative labels are critical** for robust open-vocabulary OOD detection. Each specific label contributes a localized semantic anchor and a distinct

visual prototype, enriching the negative space used for comparison. When label granularity is reduced, semantic coverage shrinks, increasing the chance that semantically adjacent OOD samples are misclassified as ID. This observation aligns with our weak-dependence analysis in Section C.2, where diverse yet weakly correlated signals jointly reduce detection variance. Practically, this motivates maintaining a sufficiently large and fine-grained OOD label bank for synthetic-exemplar-based zero-shot detection.

## G.10    Comparison with LMD on CIFAR

We further compare DualCnst (training-free) with LMD, a training-based diffusion reconstruction approach, on CIFAR benchmarks using ROC-AUC as the evaluation metric. Results are summarized in Table 36.

Table 36: ROC-AUC on CIFAR datasets. Bold indicates the higher ROC-AUC for each ID/OOD configuration.

| Method | ID | OOD | ROC-AUC↑ |
|---|---|---|---|
| LMD | CIFAR10 | CIFAR100 | 0.607 |
| DualCnst (ours) | CIFAR10 | CIFAR100 | **0.900** |
| LMD | CIFAR100 | CIFAR10 | 0.568 |
| DualCnst (ours) | CIFAR100 | CIFAR10 | **0.661** |

DualCnst substantially outperforms LMD on these CIFAR OOD settings, with a particularly large gain of +29.3 percentage points when CIFAR10 is ID and CIFAR100 is OOD (Table 36). This suggests that DualCnst's explicit multimodal consistency—linking text-derived semantics and visual exemplars—produces stronger discriminative cues than the implicit generative reconstruction objective used in LMD. Moreover, since DualCnst requires *no training or fine-tuning*, its superior performance demonstrates the effectiveness of leveraging pre-trained multimodal models for zero-shot OOD detection. In contrast, LMD's diffusion reconstruction tends to blur semantic boundaries and overfit to ID image priors, reducing generalization to unseen categories.

## G.11    Controlled Semantic Mixing: Dog–House Hybrids

To stress-test the visual branch, we synthesize controlled hybrids that mix ID semantics (dog) and OOD semantics (house). Table 37 reports average text and image similarities to the canonical ID dog exemplar.

Table 37: Similarity statistics under dog–house composition. Values are average cosine similarities to the ID "dog" exemplar.

| Setting | Text Sim (ID dog) | Image Sim (ID dog) |
|---|---|---|
| ID dog | 0.245 | 0.575 |
| OOD house | 0.178 | 0.461 |
| house with a dog | 0.235 | 0.493 |

**Interpretation.**    The hybrid "house with a dog" shows an intermediate behaviour: its text similarity to the 'dog' label (0.235) is close to the pure ID 'dog' (0.245), reflecting semantic overlap, while its image similarity (0.493) is substantially lower than the ID dog (0.575) but higher than the pure house (0.461). This separation—higher text similarity but reduced visual similarity—illustrates why multimodal fusion helps: text-only scoring risks classifying hybrids as ID (semantic match), whereas adding image-space constraints lowers their aggregated score and allows the detector to flag them as OOD. In other words, fusing weakly dependent cues recovers robustness to semantic mixing and reduces false-positive acceptance of visually atypical but semantically related inputs.

### G.12 Score Function

We present the specific form of the score function designed in the ablation study. They are $S_{\text{MAX}}$, $S_{\text{Energy}}$ and $S_{\text{MaxLogit}}$. Firstly, we review the definition of the fused visual-text cosine similarity $\tilde{s}$ as:

$$\tilde{s}_i(\mathbf{x}) = \alpha \cdot s_{i,\text{img}}(\mathbf{x}) + (1 - \alpha) \cdot s_{i,\text{text}}(\mathbf{x}) \tag{9}$$

where

$$s_{i,\text{img}}(\mathbf{x}) = \sum_{l=1}^{L} w_l \cdot s_{i,\text{img}}^{(l)}(\mathbf{x})$$

with

$$s_{i,\text{img}}^{(l)}(\mathbf{x}) = \frac{\mathcal{I}^{(l)}(\mathbf{x}) \cdot \mathcal{I}^{(l)}(\bar{\mathbf{x}}_i)}{\|\mathcal{I}^{(l)}(\mathbf{x})\| \cdot \|\mathcal{I}^{(l)}(\bar{\mathbf{x}}_i)\|}, \quad \bar{\mathbf{x}}_i \in \bar{\mathcal{X}} \tag{10}$$

and

$$s_{i,\text{text}}(\mathbf{x}) = \frac{\mathcal{I}(\mathbf{x}) \cdot \mathcal{T}(\mathbf{t}_i)}{\|\mathcal{I}(\mathbf{x})\| \cdot \|\mathcal{T}(\mathbf{t}_i)\|} \tag{11}$$

The specific form of $S_{\text{MAX}}$ is as follows:

$$S_{\text{MAX}}(\mathbf{x}; \mathcal{Y}^{\text{id}} \cup \mathcal{Y}^{\text{ood}}, \bar{\mathcal{X}}, \mathcal{T}, \mathcal{I}) = \begin{cases} \frac{1}{K}, & \max\limits_{i \in [1,K]} \tilde{s}_i < \max\limits_{j \in [K+1,K+M]} \tilde{s}_j, \\ \max\limits_{i \in [1,K]} \frac{e^{\tilde{s}_i(\mathbf{x})}}{\sum_{j=1}^{K} e^{\tilde{s}_j(\mathbf{x})}}, & \max\limits_{i \in [1,K]} \tilde{s}_i \geq \max\limits_{j \in [K+1,K+M]} \tilde{s}_j. \end{cases} \tag{12}$$

$S_{\text{MAX}}$ indicates that if the $\tilde{s}_j$ ($j \in [K+1, K+M]$) of an input sample is larger than the $\tilde{s}_i$ ($i \in [1, K]$), this sample is recognized to be an OOD sample. This implies that the maximum similarity observed between the input sample and any OOD visual-text similarity exceeds the similarity between the input sample and any ID visual-text similarity. Otherwise, the input sample is evaluated based on the maximum softmax probability.

Similarly, $S_{\text{Energy}}$ and $S_{\text{MaxLogit}}$ are modifications of the Energy and MaxLogit metrics, respectively, incorporating visual-text similarity into their secondary components.

$$S_{\text{Energy}}(\mathbf{x}; \mathcal{Y}^{\text{id}} \cup \mathcal{Y}^{\text{ood}}, \bar{\mathcal{X}}, \mathcal{T}, \mathcal{I}) = -T \left( \log \sum_{i=1}^{K} e^{\tilde{f}_i(\mathbf{x})/T} - \log \sum_{j=K+1}^{K+M} e^{\tilde{f}_j(\mathbf{x})/T} \right), \tag{13}$$

$$S_{\text{MaxLogit}}(\mathbf{x}; \mathcal{Y}^{\text{id}} \cup \mathcal{Y}^{\text{ood}}, \bar{\mathcal{X}}, \mathcal{T}, \mathcal{I}) = \max\limits_{i \in [1,K]} \tilde{s}_i(\mathbf{x}) - \max\limits_{j \in [K+1,K+M]} \tilde{s}_j(\mathbf{x}). \tag{14}$$

Table 34 presents the detailed experimental results on ImageNet-1k (ID).

