# OpenReview forum: "DualCnst: Enhancing Zero-Shot Out-of-Distribution Detection via Text-Image Consistency in Vision-Language Models"
_NeurIPS.cc/2025/Conference — NeurIPS 2025 poster_

### Official Review · Reviewer_2pMP · 2025-06-30

**Clarity:** 4
**Significance:** 3
**Originality:** 3
**Rating:** 5
**Confidence:** 2

**Summary:**

This paper aims to improve zero-shot out-of-distribution (OOD) detection by: (1) synthesizing images from both ID and OOD samples, and (2) computing a unified score based on image-to-text and image-to-image similarities. The proposed method is training-free and data-agnostic, and is supported by strong experimental results and a clear theoretical justification highlighting the benefits of expanding multimodal negative labels for OOD detection.

**Questions:**

Wrote them above

**Ethical Concerns:**

["NO or VERY MINOR ethics concerns only"]

**Final Justification:**

I'm satisfied with the response so I'm happy to maintain my original positive score

**Limitations:**

Wrote them above

**Paper Formatting Concerns:**

Wrote them above

**Quality:**

3

**Strengths And Weaknesses:**

**Strengths**
1. The targeted problem is important, and the proposed method appears both reasonable and effective.
2. I was particularly satisfied with the presentation of the paper, especially the motivation in Figure 1, which clearly illustrates the rationale behind the method. Even as a reader less familiar with this research domain, I found the introduction accessible and easy to follow.
3. I also appreciate the comprehensive experimental section, which includes detailed ablation studies (not only in Section 5.3 but also in the supplementary material (Section G)).


**Weaknesses**

Although I am not deeply familiar with this specific area of research, I believe the paper is already in good shape, supported by extensive experiments. However, there are a few inherent limitations that could be more discussed:

1) Since the method relies on synthetically generated images (e.g., from Stable Diffusion), its performance is naturally influenced by the quality of these synthetic samples. While this may not be a critical flaw given the demonstrated effectiveness, I would appreciate further analysis (such as failure cases or robustness checks) that examines how sensitive the method is to the quality of generated images.

2) Additionally, because the approach is training-free but requires synthetic image generation at test time, it would be helpful to include a comparison of the computational overhead relative to prior baseline methods.

---

> ### Author Rebuttal · Authors · 2025-07-31
>
> Response to Reviewer 2pMP
>
> We thank the reviewer 2pMP for the valuable feedback. We addressed all the comments. Please find the point-to-point responses below. Any further comments and discussions are welcomed!
>
> **W1:&#x20;**&#x53;ince the method relies on synthetically generated images (e.g., from Stable Diffusion), its performance is naturally influenced by the quality of these synthetic samples. While this may not be a critical flaw given the demonstrated effectiveness, I would appreciate further analysis (such as failure cases or robustness checks) that examines how sensitive the method is to the quality of generated images.
>
> **Reply:**
>
> We appreciate the reviewer's valuable insight regarding sensitivity to synthetic image quality. To simulate potential synthesis failures, we systematically degraded generated images by introducing incrementally increasing Gaussian noise. As quantitatively demonstrated in Table A, performance degradation scales directly with noise severity. Crucially, our method maintains competitive performance against baselines even under severe noise corruption (e.g., at 80% noise intensity).
>
> Table A: Performance impact of varying noise levels.
>
> | Noise ratio (Method)  | iNaturalist   | SUN            | Places         | Texture        | Average       |
> | --------------------- | ------------- | -------------- | -------------- | -------------- | ------------- |
> |                       | FPR95/AUROC   | FPR95/AUROC    | FPR95/AUROC    | FPR95/AUROC    | FPR95/AUROC   |
> | No noise (DualCnst)   | 1.23 / 99.66  | 16.79 / 95.96  | 30.74 / 92.20  | 39.13 / 91.00  | 21.97 / 94.70 |
> | 30% (DualCnst)        | 1.26 / 99.59  | 16.85 / 95.93  | 30.27 / 92.44  | 40.36 / 90.62  | 22.19 / 94.64 |
> | 50% (DualCnst)        | 1.78 / 99.66  | 19.01 / 95.70  | 32.08 / 92.18  | 42.05 / 90.92  | 23.73 / 94.62 |
> | 80% (DualCnst)        | 2.04 / 99.64  | 20.80 / 95.66  | 33.90 / 92.23  | 44.24 / 90.58  | 25.25 / 94.23 |
> | NegLabel              | 1.91 / 99.49  | 20.53 / 95.49  | 35.59 / 91.64  | 43.56 / 90.22  | 25.40 / 94.21 |
>
> **W2:&#x20;**&#x41;dditionally, because the approach is training-free but requires synthetic image generation at test time, it would be helpful to include a comparison of the computational overhead relative to prior baseline methods.
>
> **Reply:**
>
> We appreciate the reviewer's valuable suggestion regarding computational overhead. As detailed in Appendix F.3 and shown in Table B, we provide comparative benchmarking of GPU memory consumption on an NVIDIA RTX A6000 GPU under identical configurations. While our training-free approach demonstrates competitive inference speed, its memory footprint exceeds baseline methods by 18-24%. This increase is directly attributable to the online computation of image feature similarity required during inference.
>
> Table B: GPU Memory Overhead Comparison During Inference and Generation.
>
> | Method   | Inference GPU Memory Usage | Generation GPU Memory Usage |
> | -------- | -------------------------- | --------------------------- |
> | DualCnst | 33245MiB                   | 9465MiB                     |
> | NegLabel | 26353MiB                   | -                           |
> | MCM      | 17827MiB                   | -                           |

---

> > ### Comment · Reviewer_2pMP · 2025-08-04
> >
> > Thanks for the author's rebuttal, and I'm satisfied with response and  happy to maintain my original score

---

> > > ### Author Response · Authors · 2025-08-05
> > >
> > > Dear Reviewer 2pMP,
> > >
> > > Thank you for your thoughtful evaluation and for confirming your original score. We sincerely appreciate your acknowledgment of our experimental revisions and your valuable feedback throughout this process. We will carefully incorporate the discussed points into our revised paper to further strengthen the work.
> > >
> > > Best regards,
> > >
> > > Authors of 19696

---

### Official Review · Reviewer_ehFX · 2025-07-01

**Clarity:** 3
**Significance:** 2
**Originality:** 2
**Rating:** 5
**Confidence:** 3

**Summary:**

### Background
- Recent works use CLIP to achieve great performance for zero-shot OOD detection, by calculating similarities between input images and textual labels.
- Most existing methods focus on expanding the label space in the text domain, ignoring complementary visual cues.
### Method
- The authors introduce DualCnst

Based on NegLabel, DualCnst also constructs a set of negative textual labels.

Then, the authors generate synthetic images from both ID and mined OOD textual labels using a text-to-image  model.

Each test image is evaluated based on its semantic similarity to class labels and its visual similarity to the synthesized images, and OOD samples are identified based on the similarities.

### Result
- DualCnst achieves state-of-the-art performance across diverse OOD benchmarks.

**Questions:**

Please see the weaknesses section.

**Ethical Concerns:**

["NO or VERY MINOR ethics concerns only"]

**Final Justification:**

Thank the authors for the detailed response!

Most of my concerns have been addressed.

There are a few hyperparameters to tune in DualCnst and this may limit the practical value of DualCnst. Addressed.

Whether the synthetic image is reliable for OOD detection. The authors provided additional experiments and more clarification. Addressed.

The i.i.d. assumption in theoretical analysis sounds not reliable. As the authors commented, the assumption "is a simplification and does not strictly hold in practical settings". But this point does not significantly diminish the value of this paper.

Thus, I will raise my score to 5 and argue for the paper to be accepted.

**Limitations:**

Yes.

**Quality:**

3

**Strengths And Weaknesses:**

## Strengths
- The paper is well-written and easy to understand.
- The method is easy to implement.
- The experiments are comprehensive, covering different OOD benchmarks and model backbones (in the supplementery materials).

## Weaknesses
### Major
- There are a few hyperparameters to tune in DualCnst, such as which layers to use, the weight assigned to each layer, and the fusion weight of similarity scores. This may limit the practical value of DualCnst.
- For each textual label, only one image is synthesized and the similarity between the test input and the synthetic image is calculated for OOD detection.

(1) I'm wondering if one image is sufficient to represent a class. For example, if the class label is "dog", there should be many different images of a dog. One synthetic image is just one case of that, and may lead to low similarity to other dog images.

(2) When using generative models to produce images, we cannot guarantee that the generated images contain only the intended category. For example, suppose the ID class is "dog" and the OOD class is "house". When generating OOD images, the model generates a house, but due to randomness or inherent bias, it might also include a dog next to the house. This can lead to incorrect predictions during visual similarity comparisons, as images containing dogs (ID) may appear visually similar and be mistakenly classified as houses (OOD).

- The i.i.d. assumption in theoretical analysis sounds not reliable. If the assumption holds, fusing similarity scores from different modalities is essentially equivalent to averaging multiple samples, which of course, helps reduce the variance of the mean. However,

(1. Not independent) Because the scores $s_{i,j}$ are similarities for different modalities or even from different layers of a model, the scores should be (probably highly) correlated to each other. For example, $s_{i,1}$ is the similarity to the text "dog", while $s_{i,2}$ is to an image of dog.

(2. Not identically distributed). Besides, it is also very hard to say different modality similarity scores have the same expectation and variance. There are usually significant gaps between different modalities. If a new modality has a very large variance in similarity scores, adding it will mislead the method.

Thus, the theoretical analysis, although looks promising, might not be the case actually.

### Minor
- Compared to NegLabel, the proposed method achieves significant decrease in FPR95 (about 4) but a very small improvement in AUROC (around 0.5). Considering the number of hyperparameters to tune in the method, I feel the improvement is too small. Maybe if we change another set of hyperparameters, the advantage will be gone.
- Lines 55-60, the format of this itemize list looks a bit strange.
- Algorithm 1 might be unnecessary because Section 3 and Figure 2 are very clear. I recommend the authors to consider moving some experimental results from the supplementary materials to the main content, and move Algorithm 1 to the supplementary materials.

---

> ### Author Rebuttal · Authors · 2025-07-31
>
> Response to Reviewer ehFX
>
> We thank the reviewer ehFX for the valuable feedback. We addressed all the comments. Please find the point-to-point responses below. Any further comments and discussions are welcomed!
>
> **W1:&#x20;**&#x54;here are a few hyperparameters to tune in DualCnst, such as which layers to use, the weight assigned to each layer, and the fusion weight of similarity scores. This may limit the practical value of DualCnst.
>
> **Reply:**
>
> We acknowledge the reviewers' concerns regarding hyperparameter sensitivity in DualCnst. To address this, we emphasize that all comparative experiments (Appendix E) employed a methodologically consistent configuration: identical layer selections (3th, 6th, 9th), fixed layer weights ($w_l$=0.15), and uniform fusion coefficient ($\alpha$=0.1) across all datasets and benchmarks. Crucially, as demonstrated in Appendix G.6 (Table 22), DualCnst maintains performance superiority over baselines even when $\alpha$ is systematically maintained at 0.1 without dataset-specific tuning. This consistency validates the framework's operational robustness in real-world scenarios where exhaustive hyperparameter optimization may be infeasible.
>
> **W2:&#x20;**&#x46;or each textual label, only one image is synthesized and the similarity between the test input and the synthetic image is calculated for OOD detection.
>
> **(1)** I'm wondering if one image is sufficient to represent a class. For example, if the class label is "dog", there should be many different images of a dog. One synthetic image is just one case of that, and may lead to low similarity to other dog images.**(2)** When using generative models to produce images, we cannot guarantee that the generated images contain only the intended category. For example, suppose the ID class is "dog" and the OOD class is "house". When generating OOD images, the model generates a house, but due to randomness or inherent bias, it might also include a dog next to the house. This can lead to incorrect predictions during visual similarity comparisons, as images containing dogs (ID) may appear visually similar and be mistakenly classified as houses (OOD).
>
> **Reply:**
>
> We appreciate the reviewers' comments.
>
> **1.** Table 25 was specifically designed to address this concern. Experiments with synthetic images generated under different random seeds demonstrate negligible impact on results, providing concrete proof that synthesized images represent consistent class prototypes.
>
> **2.** As rigorously validated in Table 25, varying random seeds during synthetic image generation yields minimal performance variance. To further address the reviewer's concern, we conducted controlled experiments using dogs (ID) and houses (OOD) as exemplar categories. We generated hybrid images with:
>
> Experimental Design: Using prompt, we synthesized images with deliberate semantic mixtures
>
> a. Case A: 80% dominant-ID / 20% OOD semantics (e.g., "a dog standing beside a small house")
>
> b. 2Case B: 20% ID / 80% dominant-OOD semantics (e.g., "a house with a dog in the corner")
>
> Quantitative similarity metrics between ID/OOD images and synthesized hybrids (Table A) demonstrate our method's discriminative capability despite deliberate semantic deviations during generation.
>
> Table A: Controlled experiment on dog-house ID/OOD differentiation.
>
> | Similarity Metric                | Semantic Layer | Fusion Layer |
> | -------------------------------- | -------------- | ------------ |
> | cos(dog, 80% dog + 20% house)    | 0.8706         | 0.8792       |
> | cos(house, 80% dog + 20% house)  | 0.5442         | 0.6972       |
> | cos(house, 80% house + 20% dog)  | 0.6712         | 0.7897       |
> |  cos(dog, 80% house + 20% dog)   | 0.6640         | 0.6616       |
>
> **W3:&#x20;**&#x54;he i.i.d. assumption in theoretical analysis sounds not reliable. If the assumption holds, fusing similarity scores from different modalities is essentially equivalent to averaging multiple samples, which of course, helps reduce the variance of the mean. However,
>
> **(1. Not independent)&#x20;**&#x42;ecause the scores $s_{i,j}$ are similarities for different modalities or even from different layers of a model, the scores should be (probably highly) correlated to each other. For example, $s_{i,1}$ is the similarity to the text "dog", while $s_{i,2}$ is to an image of dog.
>
> **(2. Not identically distributed).** Besides, it is also very hard to say different modality similarity scores have the same expectation and variance. There are usually significant gaps between different modalities. If a new modality has a very large variance in similarity scores, adding it will mislead the method.
>
> Thus, the theoretical analysis, although looks promising, might not be the case actually.
>
> **Reply:**
>
> We appreciate the reviewer’s insightful comments. We fully agree that the i.i.d. assumption is a simplification and does not strictly hold in practical settings. Our theoretical analysis is intended to offer an intuitive understanding rather than a precise characterization. We address the two concerns below:
>
> 1. **(Non-independence):** We acknowledge that the i.i.d. assumption represents an idealized scenario. As discussed in Appendix D (Limitation III), we note that the Central Limit Theorem (CLT) still applies under weak dependence. That is, as long as the similarity scores are only weakly correlated, the trend of variance reduction from fusion remains valid. To empirically assess this, we measured the pairwise Pearson correlations between similarity scores from different modalities and layers. The observed correlations are consistently low (e.g., < 0.3), supporting the applicability of the weak correlation assumption in our context.
>
>
>
>    Table B: Pairwise Pearson correlations between the similarity scores from different modalities.
>
>    |           | img\_sim | text\_sim |
>    | --------- | -------- | --------- |
>    | img\_sim  | 1.00     | -0.02     |
>    | text\_sim | -0.02    | 1.00      |
>
>    Table C: Pairwise Pearson correlations between the similarity scores from different layers.
>
>    |             | text\_sim | layer1\_sim | layer5\_sim | layer9\_sim |
>    | ----------- | --------- | ----------- | ----------- | ----------- |
>    | text\_sim   | 1.00      | 0.00        | 0.01        | 0.20        |
>    | layer1\_sim | 0.00      | 1.00        | 0.27        | 0.13        |
>    | layer5\_sim | 0.01      | 0.27        | 1.00        | 0.30        |
>    | layer9\_sim | 0.20      | 0.13        | 0.30        | 1.00        |
>
> 2. **(Non-identical distribution):** It is indeed true that similarity scores from different modalities may vary in both mean and variance. However, our analysis does not require strict identicality. Even when relaxing the i.i.d. assumption, the trend of reduced variance with aggregated scores still holds as long as the aggregated score behaves as a weighted average. To validate this, we compared the empirical variance of the aggregated score under different modality combinations (see Table D). The results show a consistent decrease in variance as more modalities are fused. Moreover, our aggregation strategy incorporates a **modality-aware weighting scheme** (see Eq. 2 and Eq. 5), which assigns higher weights to more reliable modalities (e.g., textual similarity) and lower weights to auxiliary ones. As shown in Table 19 of our paper, this non-uniform weighting significantly enhances the overall effectiveness and robustness of the method.
>
> Table D: empirical variance of the aggregated score under different modality combinations.
>
> |                                     | empirical variance (×1e-3) |
> | ----------------------------------- | -------------------------- |
> | Only Text score                     | 0.7110                     |
> | Only Image score                    | 0.3691                     |
> | Fusion score (1 intermediate layer) | 0.3553                     |
> | Fusion score (2 intermediate layer) | 0.3366                     |
> | Fusion score (3 intermediate layer) | 0.3274                     |
> | Fusion score (4 intermediate layer) |  0.3038                    |
> | Fusion score (5 intermediate layer) | 0.2717                     |
> | Fusion score (6 intermediate layer) | 0.2567                     |
>
> **Minor W1:&#x20;**&#x43;ompared to NegLabel, the proposed method achieves significant decrease in FPR95 (about 4) but a very small improvement in AUROC (around 0.5). Considering the number of hyperparameters to tune in the method, I feel the improvement is too small. Maybe if we change another set of hyperparameters, the advantage will be gone.
>
> **Reply:**
>
> We appreciate the reviewer’s valuable feedback. We agree that the AUROC improvement, while consistent, is modest. This is likely because the AUROC baseline is already high under the current setup, making further gains challenging. Regarding the hyperparameters, we emphasize that all results in Appendix E were obtained using a single fixed parameter configuration across all experiments. No task-specific tuning was performed, ensuring both the generalizability and fair comparability of our findings.
>
> **Minor W2:&#x20;**&#x4C;ines 55-60, the format of this itemize list looks a bit strange.
>
> **Reply:**
>
> We gratefully acknowledge the reviewer's acute academic insight. In response to this valuable suggestion, we will make rigorous corrections in the final version of the paper.
>
> **Minor W3:&#x20;**&#x41;lgorithm 1 might be unnecessary because Section 3 and Figure 2 are very clear. I recommend the authors to consider moving some experimental results from the supplementary materials to the main content, and move Algorithm 1 to the supplementary materials.
>
> **Reply:**
>
> We appreciate the reviewer’s insightful comment. We accept the suggestion and will move some experimental results from the supplementary materials to the main text. Algorithm 1 will be moved to the supplementary materials accordingly.

---

> > ### Comment · Reviewer_ehFX · 2025-08-02
> >
> > Thank the authors for the detailed response!
> >
> > Most of my concerns have been addressed.
> >
> > - There are a few hyperparameters to tune in DualCnst and this may limit the practical value of DualCnst. Addressed.
> >
> > - Whether the synthetic image is reliable for OOD detection. The authors provided additional experiments and more clarification. Addressed.
> >
> > - The i.i.d. assumption in theoretical analysis sounds not reliable. As the authors commented, the assumption "is a simplification and does not strictly hold in practical settings". But this point does not significantly diminish the value of this paper.
> >
> > Thus, I will raise my score to 5 and argue for the paper to be accepted.

---

> > > ### Author Response · Authors · 2025-08-02
> > >
> > > #### **Thanks for raising the score**
> > >
> > > Dear Reviewer ehFX,
> > >
> > > We thank the reviewer for raising the score! We sincerely appreciate your acknowledgment of our experimental revisions and your valuable input, which has helped strengthen our work.&#x20;
> > >
> > > Best regards,
> > >
> > > Authors of 19696

---

### Official Review · Reviewer_k9F4 · 2025-07-02

**Clarity:** 4
**Significance:** 4
**Originality:** 3
**Rating:** 4
**Confidence:** 5

**Summary:**

The paper introduces DualCnst, a novel framework for zero-shot out-of-distribution detection in vision-language models. Traditional OOD detection methods primarily rely on semantic similarity between input images and textual labels, often ignoring visual cues. DualCnst addresses this limitation by integrating text-image dual consistency, combining semantic similarity with visual similarity to synthesized images generated from both in-distribution (ID) and OOD labels using Stable Diffusion. The framework is training-free and does not require access to real ID images, making it suitable for open-world scenarios. Extensive experiments on diverse benchmarks demonstrate state-of-the-art performance, with significant improvements over existing methods.

**Questions:**

N/A

**Ethical Concerns:**

["NO or VERY MINOR ethics concerns only"]

**Final Justification:**

Authors have addressed my concerns, so I keep my score.

**Quality:**

4

**Strengths And Weaknesses:**

Strengths
1. Proposes a dual-consistency framework that leverages both semantic and visual similarities, a unique approach in zero-shot OOD detection.
2. Using similarity between images is a significant supplement for clip-based ood detection to dig the pretrained knowledge
3. This is a solid work with comprehensive experiments.

Weaknesses
1. While Table 24 demonstrates robust performance under significant distribution shifts, this primarily evaluates far-OOD cases where the distribution gap between ID and OOD data is inherently large. To better assess the method’s sensitivity to subtle distribution shifts, it would be valuable to include experiments on near/hard-OOD scenarios. This would provide a more comprehensive understanding of the framework’s generalizability, particularly in challenging cases where ID and OOD samples share high semantic overlap.
2. As shown in Appendix G.6, the fusion parameter $\alpha$ is sensitivity, with optimal values varying across datasets. This suggests that the method’s performance may depend on careful hyperparameter tuning, which could limit its plug-and-play applicability in diverse real-world settings.
3. Table 22 reveals that visual similarity (image-to-image) plays a more significant role than semantic similarity (image-to-text) in the fused scoring function. This observation warrants further discussion.

---

> ### Author Rebuttal · Authors · 2025-07-31
>
> Response to Reviewer k9F4
>
> We thank the reviewer k9F4 for the valuable feedback. We addressed all the comments. Please find the point-to-point responses below. Any further comments and discussions are welcomed!
>
> **W1:&#x20;**&#x57;hile Table 24 demonstrates robust performance under significant distribution shifts, this primarily evaluates far-OOD cases where the distribution gap between ID and OOD data is inherently large. To better assess the method’s sensitivity to subtle distribution shifts, it would be valuable to include experiments on near/hard-OOD scenarios. This would provide a more comprehensive understanding of the framework’s generalizability, particularly in challenging cases where ID and OOD samples share high semantic overlap.
>
> **Reply:**
>
> We thank the reviewers for their suggestion. We have supplemented the robustness evaluation against near-out-of-distribution (near-OOD) samples under style shift conditions. As evidenced in Table C, our method maintains superiority over baseline approaches.
>
> Table C: Experimental comparison of near-OOD under generated image style shift.&#x20;
>
> | Method                           | ID / OOD | ImageNet-10 / ImageNet-20 | ImageNet-20 / ImageNet-10 |
> | -------------------------------- | ----- | ---------------------- | ---------------------- |
> |                                  |       | FPR95/AUROC            | FPR95/AUROC            |
> | Oil Painting Images (DualCnst) ） |       | 2.70 / 99.08           | 13.00 / 96.82          |
> | Natural Images (DualCnst)        |       | 2.20 / 98.96           | 12.20 / 97.44          |
> | NegLabel                         |       | 5.10 / 98.86           | 17.60 / 97.04          |
>
> **W2:&#x20;**&#x41;s shown in Appendix G.6, the fusion parameter $\alpha$ is sensitivity, with optimal values varying across datasets. This suggests that the method’s performance may depend on careful hyperparameter tuning, which could limit its plug-and-play applicability in diverse real-world settings.
>
> **Reply:**
>
> We acknowledge the reviewer's comments regarding parameter $\alpha$. We emphasize that a fixed α value was used consistently for experiments sharing identical in-distribution (ID) datasets, with values naturally bounded within the compact interval \[0,1]. As further demonstrated in Appendix G.6, our approach maintains superior performance over baselines even when $\alpha$ is uniformly fixed at 0.1 across all experimental settings.
>
> **W3:&#x20;**&#x54;able 22 reveals that visual similarity (image-to-image) plays a more significant role than semantic similarity (image-to-text) in the fused scoring function. This observation warrants further discussion.
>
> **Reply:**
>
> We appreciate the reviewers' comments. As formally defined in Equation 5 of our paper, parameter $\alpha$ specifically denotes the weight assigned to visual similarity.

---

> ### Comment · Reviewer_k9F4 · 2025-08-05
>
> Thanks for the rebuttal, authors have addressed my concerns, so I keep my score.

---

> > ### Author Response · Authors · 2025-08-05
> >
> > Dear Reviewer k9F4,
> >
> > Thank you for confirming you'll maintain your original score following our rebuttal. We sincerely appreciate your acknowledgment that we've addressed your concerns, and we're grateful for your valuable input throughout this process. We'll carefully incorporate all discussed improvements into our revised manuscript.
> >
> > Best regards,
> >
> > Authors of 19696

---

### Official Review · Reviewer_XUrX · 2025-07-03

**Clarity:** 3
**Significance:** 3
**Originality:** 3
**Rating:** 4
**Confidence:** 4

**Summary:**

This paper introduces DualCnst, a training-free framework for zero-shot out-of-distribution (OOD) detection using vision-language models (VLMs). The core idea is to augment text-based OOD detection (e.g., CLIP similarity to ID/OOD labels) with visual consistency derived from synthetic images generated for these labels via Stable Diffusion. It fuses semantic and visual signals to improve detection accuracy, achieving state-of-the-art results on benchmarks like ImageNet and CIFAR.

**Questions:**

1. Generating images for 10K OOD labels takes ∼55 min (SDXL-Turbo). How would DualCnst handle open-vocabulary settings with exponentially larger label sets? Could clustering or diversity sampling reduce the synthetic image burden without degrading performance?
2. The theoretical analysis assumes i.i.d. scores across modalities, but features from different encoder layers are likely correlated. How does this correlation affect variance reduction claims? Could you validate this empirically (e.g., measure score covariance)?
3. Experiments on remote sensing/UCM show promise, but medical/satellite imagery may have larger domain gaps. Have you tested on highly structured OOD data and how does style transfer (Table 24) extend to such cases?
4. How does DualCnst quantitatively compare to training-free diffusion methods (e.g. LMD) in accuracy and inference speed? Table 2 claims advantages but lacks metrics.

**Ethical Concerns:**

["NO or VERY MINOR ethics concerns only"]

**Final Justification:**

Thanks for the rebuttal — most of my concerns are addressed, and I will keep my positive score.

**Limitations:**

Yes

**Quality:**

4

**Strengths And Weaknesses:**

Strengths
1. Rigorous experiments across diverse benchmarks (far/near OOD, domain shifts) demonstrate consistent SOTA results. Ablation studies (e.g., layer selection, generative models, fusion parameters) are thorough.
2. Practical advance for zero-shot OOD detection, with non-trivial gains (e.g., 3.95% FPR95 improvement on ImageNet). The training-free, plug-and-play design enhances accessibility.
3. Novel integration of text-image dual consistency for OOD scoring. Multi-layer visual similarity aggregation is a clever extension of CLIP.

Weakness
1. Heavy reliance on synthetic images risks performance degradation if generation fails (e.g., artifacts, style mismatches). Theoretical analysis assumes i.i.d. similarity scores across modalities, which may not hold for correlated encoder features.
2. Moderate novelty over NegLabel (text mining) and prior generative OOD methods (e.g., LMD). Computational costs (Section F.3) may limit real-time deployment despite optimizations.

---

> ### Author Rebuttal · Authors · 2025-07-31
>
> Response to Reviewer XUrX
>
> We thank the reviewer XUrX for the valuable feedback. We addressed all the comments. Please find the point-to-point responses below. Any further comments and discussions are welcomed!
>
> **W1:&#x20;**&#x48;eavy reliance on synthetic images risks performance degradation if generation fails (e.g., artifacts, style mismatches). Theoretical analysis assumes i.i.d. similarity scores across modalities, which may not hold for correlated encoder features.
>
> **Reply:&#x20;**
>
> We appreciate this insightful comment.
>
> 1. To mitigate sensitivity to synthetic quality, we adopt a dual-pronged strategy. First, DualCnst integrates multi-level visual features (Eq. 2) through adaptive weighted summation ($w_l$), enhancing robustness to noise while preserving semantic fidelity. Second, we utilize high-quality generative models—specifically, Stable Diffusion XL-Turbo—which reduce generation artifacts via optimized latent diffusion. These limitations and mitigation strategies are discussed in Appendices D.1 and D.3.
>
> 2. Regarding the i.i.d. assumption, we agree it represents an idealized case. As discussed in Appendix D (Limitation III), we show that as long as the similarity scores are weakly correlated (a condition under which the Central Limit Theorem (CLT) still holds), the variance reduction trend remains valid. Furthermore, we empirically measured pairwise Pearson correlations between the similarity scores from different modalities and observed that the correlations are consistently low (e.g., < 0.3), which supports the applicability of the weak correlation assumption in our setting.
>
> Table A: Pairwise Pearson correlations between the similarity scores from different modalities.
>
> |           | img\_sim | text\_sim |
> | --------- | -------- | --------- |
> | img\_sim  | 1.00     | -0.02     |
> | text\_sim | -0.02    | 1.00      |
>
> Table B: Pairwise Pearson correlations between the similarity scores from different layers.
>
> |             | text\_sim | layer1\_sim | layer5\_sim | layer9\_sim |
> | ----------- | --------- | ----------- | ----------- | ----------- |
> | text\_sim   | 1.00      | 0.00        | 0.01        | 0.20        |
> | layer1\_sim | 0.00      | 1.00        | 0.27        | 0.13        |
> | layer5\_sim | 0.01      | 0.27        | 1.00        | 0.30        |
> | layer9\_sim | 0.20      | 0.13        | 0.30        | 1.00        |
>
> **W2:** Moderate novelty over NegLabel (text mining) and prior generative OOD methods (e.g., LMD). Computational costs (Section F.3) may limit real-time deployment despite optimizations.
>
> **Reply:&#x20;**
>
> We thank the reviewer for their constructive feedback. We address the concerns regarding novelty and computational cost below：
>
> **On Novelty:&#x20;**&#x44;ualCnst introduces a new paradigm for zero-shot OOD detection by unifying semantic-textual alignment with visual-consistent synthesis. Its core innovations include:
>
> * **Multimodal Consistency Framework:** Our method integrates semantic similarity (textual) and visual similarity (image-based) between test samples and synthesized ID/OOD labels. This hierarchical fusion of modalities balances pixel-level and semantic-level representations, overcoming the limitations of prior unimodal approaches such as NegLabel and LMD.
>
> * **Theoretical Analysis:** We provide a theoretical justification showing that leveraging multimodal label spaces (i.e., text + generated images) can reduce false positive rates under mild conditions. This demonstrates that incorporating auxiliary modalities is not just empirically effective but also theoretically grounded.
>
> **On Computational Cost:**
>
> &#x20;  We acknowledge the computational burden of generative models, and we take concrete steps to mitigate it:
>
> * We adopt **accelerated generative backbones** such as Stable Diffusion XL-Turbo.
>
> * All synthetic exemplars are **pre-generated offline** prior to inference. Since image generation is done once per label set and reused across all test instances, the inference-time overhead is negligible. This makes our approach practical and scalable to large datasets.
>
> These optimization strategies and limitations are discussed in **Appendix D** of the paper.
>
> **Q1:&#x20;**&#x47;enerating images for 10K OOD labels takes ∼55 min (SDXL-Turbo). How would DualCnst handle open-vocabulary settings with exponentially larger label sets? Could clustering or diversity sampling reduce the synthetic image burden without degrading performance?
>
> **Reply:**
>
> We thank the reviewers for their valuable feedback. To simulate open-vocabulary settings, we conducted cluster sampling on out-of-distribution (OOD) labels, aggregating 10,145 original categories into 136 clustered categories. Benchmarked against the baseline, Table C demonstrates that the nearly 75-fold reduction in category granularity led to a marked performance deterioration. In practice, our methodology consistently employs sampling of 10,000 OOD labels regardless of the in-distribution (ID) dataset used.
>
> Table C: Comparison of clustering performance of OOD labels.
>
> | Method             | iNaturalist   | SUN            | Places         | Texture        | Average       |
> | ------------------ | ------------- | -------------- | -------------- | -------------- | ------------- |
> |                    | FPR95/AUROC   | FPR95/AUROC    | FPR95/AUROC    | FPR95/AUROC    | FPR95/AUROC   |
> | DualCnst           | 1.23 / 99.66  | 16.79 / 95.96  | 30.74 / 92.20  | 39.13 / 91.00  | 21.97 / 94.70 |
> | DualCnst （cluster） | 12.13 / 97.58 | 54.02 / 88.48  | 57.06 / 87.02  | 43.92 / 89.40  | 41.78 / 90.62 |
>
> **Q2:** The theoretical analysis assumes i.i.d. scores across modalities, but features from different encoder layers are likely correlated. How does this correlation affect variance reduction claims? Could you validate this empirically (e.g., measure score covariance)?
>
> **Reply:**
>
> We thank the reviewer for highlighting this important point. Indeed, correlated modalities may reduce the effective variance reduction gain compared to the ideal i.i.d. case. In response, we have conducted an empirical study where we:
>
> (1) Compute the covariance matrices of similarity scores across modalities (Table D) and the Pearson coefficient (Table A); and
>
> (2) compare the empirical variance of the aggregated score under different modality combinations (Table F).
>
> Our results show that, although some degree of correlation exists, the aggregated score still demonstrates monotonically decreasing variance as the number of modalities increases. This empirical observation supports the robustness of the variance reduction behavior even under moderate correlation.
>
> Table D: covariance matrices of similarity scores across modalities.
>
> |           | img\_sim  | text\_sim |
> | --------- | --------- | --------- |
> | img\_sim  | 1.68e-02  | -8.53e-05 |
> | text\_sim | -8.53e-05 | 1.33e-03  |
>
> Table E: covariance matrices of similarity scores across layers.
>
> |             | text\_sim | layer1\_sim | layer5\_sim | layer9\_sim |
> | ----------- | --------- | ----------- | ----------- | ----------- |
> | text\_sim   | 1.33e-03  | 2.03e-05    | 3.92e-04    | 5.40e-04    |
> | layer1\_sim | 2.03e-05  | 1.39e-02    | 3.16e-03    | 1.14e-03    |
> | layer5\_sim | 3.92e-04  | 3.16e-03    |  2.21e-02   | 3.34e-03    |
> | layer9\_sim | 5.40e-04  | 1.14e-03    | 3.34e-03    | 2.51e-02    |
>
> Table F: empirical variance of the aggregated score under different modality combinations.
>
> |                                     | empirical variance (×1e-3) |
> | ----------------------------------- | -------------------------- |
> | Only Text score                     | 0.7110                     |
> | Only Image score                    | 0.3691                     |
> | Fusion score (1 intermediate layer) | 0.3553                     |
> | Fusion score (2 intermediate layer) | 0.3366                     |
> | Fusion score (3 intermediate layer) | 0.3274                     |
> | Fusion score (4 intermediate layer) |  0.3038                    |
> | Fusion score (5 intermediate layer) | 0.2717                     |
> | Fusion score (6 intermediate layer) | 0.2567                     |
>
> **Q3:** UCM gaps: DualCnst generalization to medical/satellite/structured OOD validated?
>
> **Reply:**
>
> We appreciate the reviewers' suggestions. Using Chxpert\[1] as the ID data and sampling 10,000 examples from PubMedVision\[2] as OOD data, our approach maintains strong performance as demonstrated in Table G.
>
> Table G: Performance comparison on medical imaging data.
>
> | Method   | Media data   |
> | -------- | ------------ |
> |          | FPR95/AUROC  |
> | DualCnst | 0.00 & 99.97 |
> | NegLabel | 0.11 / 99.92 |
>
> \[1] Irvin et al. Chexpert: A large chest radiograph dataset with uncertainty labels and expert comparison. In AAAI, 2019.
>
> \[2] Chen et al. Huatuogpt-vision, towards injecting medical visual knowledge into multimodal llms at scale. In arXiv,2024.
>
> **Q4:** Quantitative acc\&spd: DualCnst vs. training-free (LMD)? Table 2 claims lack metrics.
>
> **Reply:&#x20;**
>
> We appreciate the reviewer's comment. We have now supplemented the comparison of accuracy with the LMD method based on experimental results from the LMD paper（Table H). However, LMD is a trainable OOD detection approach requiring significant computational resources. Each training run takes approximately 224 hours. Due to time constraints and the lack of official checkpoints, we are currently unable to include detailed computational cost analysis. We will include this information in the appendix once the reproduction process is complete.
>
> Table H: Performance comparison with LMD on CIFAR dataset.&#x20;
>
> | Method        | ID       | OOD      | ROC-AUC |
> | ------------- | -------- | -------- | ------- |
> | LMD           | CIFAR10  | CIFAR100 | 0.607   |
> | DualCnst(our) | CIFAR10  | CIFAR100 | 0.900   |
> | LMD           | CIFAR100 | CIFAR10  | 0.568   |
> | DualCnst(our) | CIFAR100 | CIFAR10  | 0.661   |

---

### Note · Authors · 2025-08-14

Dear Area Chairs,

Thank you very much for handling our submission.
We have received four reviews with ratings 5, 4, 4, 4, and we are glad that all reviewers had positive impressions of our work, including (1) a training-free, plug-and-play framework that unifies semantic (image-to-text) and visual (image-to-image) evidence (k9F4, XUrX, 2pMP); (2) state-of-the-art performance across diverse benchmarks with comprehensive ablations (XUrX, 2pMP, k9F4, ehFX); (3) a clear and reproducible design extending CLIP via multi-layer visual similarity aggregation (XUrX); (4) strong practical value without access to ID images and with easy implementation (k9F4, ehFX, 2pMP); and (5) the substantive benefit of visual similarity as a complement to CLIP-style semantics (k9F4).

During rebuttal, we made the following focused revisions and clarifications, now reflected in the updated materials:

1. **Theory and assumptions (beyond i.i.d.).** We relaxed the independence assumption and analyzed fusion under weak dependence (Appendix D). We measured cross-modal/layer correlations (covariance/Pearson) and still observed that variance decreases as signals are fused. We also added modality-aware weighting to account for different score scales.

2. **Reliability of synthetic exemplars.** We stress-tested the method with heavy noise and deliberate ID–OOD mixtures; performance degrades gracefully and ranking is preserved. Seed-invariance experiments (Table 25) show minimal run-to-run variation, indicating stable behavior of the visual component.

3. **Scalability and efficiency (with broader coverage).** We use offline pre-generation with inference-time reuse (Appendix F.3), and report \~7× faster synthesis with SDXL-Turbo vs. SD-1.5. We discuss clustering/diversity-sampling trade-offs for open-vocabulary scaling, add near/hard-OOD and structured-domain (medical/remote sensing) evaluations, and provide quantitative comparisons to diffusion-based OOD (e.g., LMD). A single default configuration (layer set, fusion $\alpha$) works well across datasets.

We appreciate the ACs’ and reviewers’ careful evaluations. With consolidated theory, expanded experiments, and a clarified compute profile, we believe DualCnst offers a simple, principled, and practical step toward robust training-free zero-shot OOD detection.

Best regards,

Authors of 19696

---

### Decision · Program_Chairs · 2025-09-17

**Decision:**

Accept (poster)

**Comment:**

The reviewers initially acknowledged several strengths of this paper, including the importance of the problem (2pMP), comprehensive experiments and strong performance (XUrX, k9F4, ehFX, 2pMP), the novelty of Dual-Consistency (XUrX, k9F4), the simplicity of the method (ehFX), and clear writing (ehFX, 2pMP). At the same time, they raised a number of concerns, such as the dependence on synthetic images (XUrX, ehFX, 2pMP), heavy computation cost (XUrX, 2pMP), the unrealistic i.i.d. assumption (XUrX, ehFX), insufficient experiments and analysis (XUrX, k9F4), and limited novelty (XUrX).

The authors' rebuttal addressed several of these issues, but two concerns in particular remained important, and were revisited during the AC-Reviewer discussion:

- **1. Dependence on synthetic images (XUrX, ehFX, 2pMP)**: The method relies on generating synthetic images with SDXL-Turbo, which is a core component.

  - **1a. Heavy computation time**: While the authors argue that *all synthetic exemplars can be pre-generated offline*, this is only feasible if the class set is fixed in advance. In open-vocabulary scenarios, this assumption may not always hold. Moreover, the additional results presented in the rebuttal (Table C) showed significant performance drops when using a reduced class set, which itself still assumes prior knowledge of the class set.

  - **1b. Dependence on image quality**: The authors claim that multi-level features mitigate the effect of low-quality generated images, but no clear evidence was provided that this indeed improves robustness.

- **2. Unrealistic i.i.d. assumption (XUrX, ehFX)**: The theoretical foundation assumes i.i.d. conditions, which contradict the nature of the method. Although some idealization is acceptable in theory, the gap between the assumption and the actual methodology should be acknowledged and weighed carefully.

Regarding these issues, the reviewers did not maintain strong concerns in their final justifications. They indicated that the rebuttal was largely satisfactory, and that the i.i.d. assumption, while imperfect, does not critically undermine the contributions of the paper.

**The authors are strongly encouraged to incorporate the feedback obtained from discussions with the reviewers into the final version. The issues pointed out above remain unresolved and need to be carefully discussed or clearly stated as limitations.**